# Influence of modifications (from AoB2015 to v0.5) in the Vegetation Optimality Model

Remko C. Nijzink[1], Jason Beringer[2], Lindsay B. Hutley[3], and Stanislaus J. Schymanski[1]

[1]Luxembourg Institute of Science and Technology, Environmental Research and Innovation, Catchment and Eco-hydrology Research Group, Belvaux, Luxembourg
[2]School of Agriculture and Environment, The University of Western Australia, Crawley, WA, Australia, 6909
[3]Research Institute for the Environment and Livelihoods, Charles Darwin University, Darwin, NT, Australia, 0909

**Correspondence:** R.C. Nijzink (remko.nijzink@list.lu)

**Abstract.**

The Vegetation Optimality Model (VOM, Schymanski et al., 2009, 2015) is an optimality-based, coupled water-vegetation model that predicts vegetation properties and behaviour based on optimality theory, rather than calibrating vegetation properties or prescribing them based on observations, as most conventional models do. Several updates to previous applications of the VOM have been made for the study in the accompanying paper of Nijzink et al. (2021), where we assess whether optimality theory can alleviate common shortcomings of conventional models, as identified in a previous model inter-comparison study along the North Australian Tropical Transect (NATT) (Whitley et al., 2016). Therefore, we assess in this technical paper how the updates to the model and input data would have affected the original results of Schymanski et al. (2015), and we implemented these changes one at a time.

The model updates included extended input data, the use of variable atmospheric $CO_2$-levels, modified soil properties, implementation of free drainage conditions, and the addition of grass rooting depths to the optimized vegetation properties. A systematic assessment of these changes was carried out by adding each individual modification to the original version of the VOM at the flux tower site of Howard Springs, Australia.

The analysis revealed that the implemented changes affected the simulation of mean annual evapo-transpiration (ET) and gross primary productivity (GPP) by no more than 20%, with the largest effects caused by the newly imposed free drainage conditions and modified soil texture. Free drainage conditions led to an underestimation of ET and GPP in comparison with the results of Schymanski et al. (2015), whereas more fine-grained soil textures increased the water storage in the soil and resulted in increased GPP. Although part of the effect of free drainage was compensated for by the updated soil texture, when combining all changes, the resulting effect on the simulated fluxes was still dominated by the effect of implementing free drainage conditions. Eventually, the relative error for the mean annual ET, in comparison with flux tower observations, changed from an 8.4% overestimation to an 10.2% underestimation, whereas the relative errors for the mean annual GPP remained similar with an overestimation that slightly reduced from 17.8% to 14.7%. The sensitivity to free drainage conditions suggests that a realistic representation of groundwater dynamics is very important for predicting ET and GPP at a tropical open-forest savanna site as investigated here. The modest changes in model outputs highlighted the robustness of the optimization approach that is central to the VOM architecture.

# 1 Introduction

Novel modelling approaches that are able to explicitly model vegetation dynamics, such as vegetation cover or root surfaces, may lead to an overall improved understanding of carbon and water flux exchanges with the atmosphere. At the same time, terrestrial biosphere models (TBMs) often rely on (remotely sensed) observations of vegetation properties, such as the leaf area index (LAI) or vegetation cover, complicating the ability of these models to make predictions for future scenarios. In addition, this makes the models rely on the quality of the data and does not enhance our understanding of the vegetation dynamics, which is highly important regarding the feedbacks between the land and the atmosphere in a changing climate. Recent model inter-comparison studies also confirm that models with explicit vegetation dynamics are needed, as Whitley et al. (2016) showed that prescribing rooting depths and the lack of dynamic representations of LAI led to highly variable performances for a selection of TBMs. Optimality theory predicts the variation and dynamics of vegetation cover, root systems, water use and carbon uptake without the need for site-specific input about vegetation properties, by optimizing these properties for a certain objective, such as maximizing the carbon gain by photosynthesis (Hikosaka, 2003; Raupach, 2005; Buckley and Roberts, 2006) or minimizing water stress (Rodriguez-Iturbe et al., 1999; Rodríguez-Iturbe et al., 1999). The theory used here is based on the premise that the Net Carbon Profit (NCP, Schymanski et al., 2007, 2008a, 2009), which is the difference between carbon assimilated by photosynthesis and carbon expended on construction and maintenance of all the plant tissues needed for photosynthesis, water uptake and storage, is an appropriate measure of plant fitness, given that assimilated carbon is a fundamental resource of plant growth, development, survival and reproduction. The theory further assumes that construction and maintenance costs of plant organ functionality are general and therefore transferable between species and sites. Hence, the costs and benefits at different sites are determined in a consistent way, leading to vegetation properties that solely depend on physical conditions, such as meteorological forcing, soils and hydrology. As a result, this leads to a systematic and consistent explanation of vegetation behaviour under different external conditions at different sites.

These optimality principles were employed in the Vegetation Optimality Model (VOM, Schymanski et al., 2009, 2015). The VOM is a coupled water-vegetation model that optimizes vegetation properties to maximize the NCP in the long-term (20-30 years) for given climate and physical properties at the site under consideration. The VOM has been previously applied by Schymanski et al. (2009) and Schymanski et al. (2015) at Howard Springs, a flux tower site in the North Australian Tropical Transect (NATT, Hutley et al., 2011). The NATT consists of multiple flux tower sites along a precipitation gradient from north to south, which allows for a more systematic testing of the VOM under different climatological circumstances. The NATT has been used previously in an intercomparison of terrestrial biosphere models (TBMs) by Whitley et al. (2016), which revealed that lacking or wrong vegetation dynamics and incorrect assumptions about rooting depths have a strong influence on the performance of state-of-the-art TBMs. Previous studies show that rooting depths vary considerably with precipitation (Schenk and Jackson, 2002) and, thus, also along the North Australian Tropical Transect (Williams et al., 1996; Ma et al., 2013). In contrast to these TBMs, the VOM predicts rooting depths and vegetation dynamics, and provides therefore a novel approach for the simulation of these savanna sites. In order to find out if this novel approach can help to overcome the shortcomings of

common TBMs, in the accompanying paper by Nijzink et al. (2021) the VOM was applied to the same sites along the NATT and systematically compared with the previous simulations presented by Whitley et al. (2016).

In order to understand if predicted optimality-based rooting depths and vegetation cover result in better simulations, Nijzink et al. (2021) ran the VOM both with predicted and prescribed rooting depths and vegetation cover while systematically comparing simulated fluxes with observations and the output of the other TBMs. For that reason, Nijzink et al. (2021) made several changes to the VOM set-up of Schymanski et al. (2015) in order to use the same input data and similar physical boundary conditions at the different sites as the TBMs in Whitley et al. (2016). In the remainder of this paper, the new set-up in Nijzink et al. (2021) will be referred to as VOM-v0.5, in contrast to VOM-AoB2015 for the set-up of Schymanski et al. (2015).

First, in the simulations by Whitley et al. (2016), all TBMs were run under the assumption of a freely draining soil column. In contrast, the VOM-AoB2015 used a hydrological schematization based on the local topography around the flux tower site (Schymanski et al., 2008b), which resulted in groundwater tables varying around 5 m below the surface. For better comparability with Whitley et al. (2016), the boundary conditions of the VOM were adjusted to resemble freely draining conditions. Assessment of the influence of this change could lead to additional insights about the influence of groundwater on the resulting carbon and water fluxes, which can be significant (York et al., 2002; Bierkens and van den Hurk, 2007; Maxwell et al., 2007).

Another modification concerns the prescribed atmospheric $CO_2$-concentrations. The VOM-AoB2015 assumed constant atmospheric $CO_2$-concentrations over the entire modelling period, whereas the VOM-v0.5 used measured $CO_2$-concentrations, which have increased considerably over the past years (Keeling et al., 2005). The previously documented influence of atmospheric $CO_2$-concentrations on the water and carbon fluxes simulated by the VOM (Schymanski et al., 2015) calls for a systematic assessment of this change.

In the VOM-AoB2015, the grass rooting depth was prescribed to a value of 1 m, arguing that only tree roots could penetrate into deeper layers due to the presence of a hard pan at Howard Springs, which not necessarily exists at the other sites along the NATT. In order to make the VOM applicable to all sites along the NATT, the grass rooting depth was not prescribed but also optimized for NCP in the VOM-v0.5.

In addition to the changes in the boundary conditions, model code and parametrization mentioned above, higher computational power, allowing for finer discretizations, and updated forcing data may also affect simulation results. In general, reproducing benchmark datasets is often seen as necessary in order to be confident about the model and the numerical implementation (Blyth et al., 2011; Abramowitz, 2012; Clark et al., 2021). Here we argue that it is also important to assess effects of individual changes one at a time, to avoid obtaining "the right results for the wrong reasons" as two errors may compensate for each other when comparing the final results with a given benchmark.

Therefore, we assess to what extent the various changes influence the VOM-results, by adding these changes one at a time to the VOM-AoB2015, for the same flux tower site in Australia, Howard Springs, as in Schymanski et al. (2009, 2015). This technical note describes the nine changes to the VOM since its last application by Schymanski et al. (2015), and how they affect the results of the VOM individually and in combination. In this way, this work should point at important model decisions and sensitivities of the VOM, as well as TBMs in general. At the same time, this work showcases a systematic evaluation of model updates and changes, which we deem necessary in model applications.

## 2 Methodology

All steps in the process, from pre- and post-processing to model runs, were done in an open science approach using the RENKU[1] platform. The workflows including code and input data can be found online[2,3]. In the following, we briefly describe the study site, the VOM, and the various modifications done in this study, compared to Schymanski et al. (2015).

### 2.1 Study site

The study site used by Schymanski et al. (2009) and Schymanski et al. (2015) is Howard Springs (AU-How), which was
100 therefore used in this analysis as well. At the same time, the flux tower site at Howard Springs provides a long record of carbon dioxide and water fluxes starting from 2001 (Beringer et al., 2016). Howard Springs is the wettest site with an average precipitation of 1747 mm/year (SILO Data Drill, Jeffrey et al., 2001, calculated for 1980-2017) along the North Australian Tropical Transect (NATT, Hutley et al., 2011), which has a strong precipitation gradient from north to south, with a mean annual precipitation around 500 mm/year at the driest site. The vegetation at Howard Springs consists of a mostly evergreen
overstorey (mainly *Eucalyptus miniata* and *Eucalyptus tetrodonta*) and an understorey dominated by annual *Sorghum* and *Heteropogon* grasses. The soils at Howard Springs are well-drained red and grey kandosols, and have a high gravel content and a sandy loam structure.

### 2.2 Vegetation Optimality Model

The Vegetation Optimality Model (VOM, Schymanski et al., 2009, 2015) is a coupled water and vegetation model, that opti-
110 mizes vegetation properties by maximizing the NCP. The model code and documentation can be found online[4,5] and version v0.5[6] of the model was used here. The processes and parameterizations are not modified in the VOM-v0.5, unless explicitly stated in Sect. 2.2.9. Therefore, more details of the VOM can be found in Schymanski et al. (2009, 2015), whereas detailed descriptions about the root processes can be found in Schymanski et al. (2008b) and the canopy processes in Schymanski et al. (2007). Nevertheless, a general description of the VOM is given below for completeness.

### 2.2.1 Water balance model

The soil is schematized as a permeable block containing an unsaturated zone and a saturated zone (see Figure 2), overlaying an impermeable bedrock with a prescribed drainage level. The model simulates a variable water table based on the vertical fluxes between horizontal soil layers and a drainage flux computed as a function of the water table elevation. The vertical fluxes between soil layers are determined using a discretization of the Buckingham-Darcy equation (Radcliffe and Rasmussen,

---

[1]https://renkulab.io/

[2]https://renkulab.io/gitlab/remko.nijzink/vomcases

[3]https://doi.org/10.5281/zenodo.5789101

[4]https://github.com/schymans/VOM

[5]https://vom.readthedocs.io

[6]https://doi.org/10.5281/zenodo.3630081

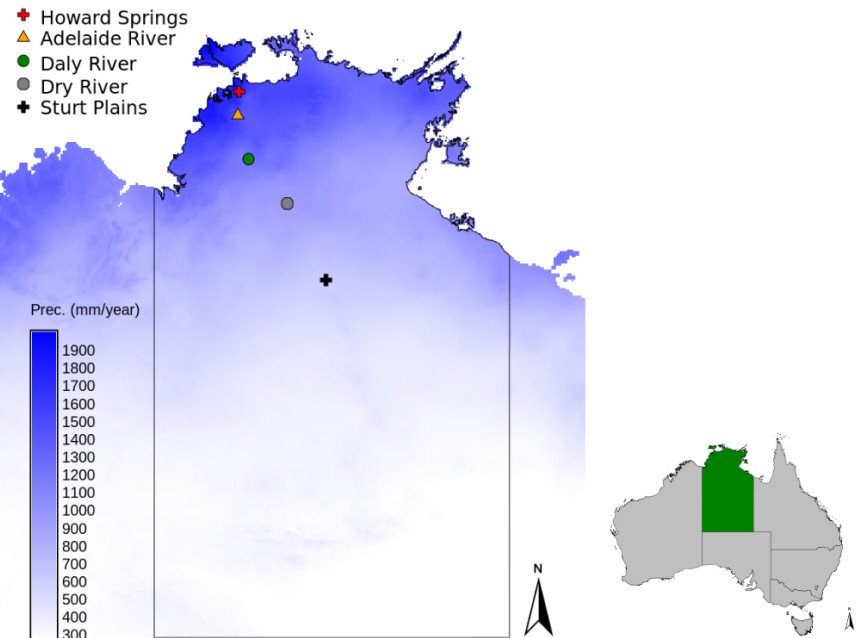

**Figure 1.** Location of the Howard Springs site, together with the other flux tower sites that are part of the North Australian Tropical Transect in the Nortern Territory of Australia, with the mean annual precipitation shown in the blue colorscale (SILO Data Drill, Jeffrey et al., 2001, calculated for 1980-2017).

2002), resulting in the 1-D Richards' equation of steady flow. The matric suction heads and unsaturated conductivities were determined with the model of van Genuchten (1980).

The hydrological parameters that determine the drainage outflow and groundwater tables are a hydrological length scale for seepage outflow, channel slope and drainage level $z_r$, based on Reggiani et al. (2000). The seepage outflow is determined by the elevation difference between groundwater table and drainage level, divided by a resistance term that uses the hydrological length scale and channel slope (Eq. 10, Schymanski et al., 2008b):

$$Q_{s,f} = K_{sat} \cdot \frac{\omega_0 \cdot (y_s - z_r)}{2 \cdot cos(\gamma_0) \cdot \Lambda_s} \tag{1}$$

with $\gamma_0$ the average slope of the seepage face (rad), $\Lambda_s$ a hydrological length scale (m), $y_s$ the groundwater table (m), $K_{sat}$ the saturated hydraulic conductivity (m s$^{-1}$), $\omega_0$ the saturated surface area fraction and $z_r$ the drainage level (m).

As illustrated in Figure 2, when precipitation falls on this soil block, it either causes immediate surface runoff or infiltrates. Once infiltrated, it can be taken up by roots and transpired, or it can evaporate at the soil surface, or move downwards until it drains away at a depth defined by the drainage level $z_r$ and total soil thickness $c_z$. In the top soil layer, soil evaporation is

**Table 1.** Characteristics of the Howard Springs site. Vegetation data from Hutley et al. (2011), Hutley (2015) and Whitley et al. (2016), with Eucalyptus (Eu.), Erythrophleum (Er.), Hetropogan (He.). Meteorological data is taken from the SILO Data Drill (Jeffrey et al., 2001) for the model periods of 1-1-1980 until 31-12-2017, with the reference crop evapotranspiration calculated according to the FAO Penman-Monteith formula (Allen et al., 1998). The ratio of the net radiation $Rn$ with the latent heat of vaporization $\lambda$ mutiplied with the precipitation $P$, is defined here as the aridity $Rn/\lambda P$. Tree cover is determined as the minimum value of the mean monthly projective cover based on fPAR-observations (Donohue et al., 2013). The maximum grass cover was found by subtracting the tree cover from the remotely sensed projective cover.

| Study Site | Howard Springs |
|---|---|
| FLUXNET ID | AU-How |
| Coordinates | 12.49S |
| | 131.35E |
| Precipitation (mm year$^{-1}$) | 1747 |
| Ref. crop evapotranspiration (mm year$^{-1}$) | 1763 |
| Aridity (-) | 1.03 |
| Net Radiation (MJ m$^{-2}$ year$^{-1}$) | 4392 |
| Mean maximum temp. [$^{o}$C] | 37.5 |
| Mean minimum temp. [$^{o}$C] | 27.4 |
| Tree cover (%) | 39.8 |
| Maximum grass cover (%) | 44.3 |
| Species | |
| Overstorey | *Eu. miniata* |
| | *Eu. tetrodonta* |
| | *Er. chlorostachys* |
| Understorey | *Sorghum spp.* |
| | *He. triticeus* |

determined as a function of the soil moisture, global radiation and vegetation cover (Eq. 18, 19, Sect. 3.3.3, Schymanski et al., 2009).

### 2.2.2 Vegetation model

The VOM schematizes the ecosystem as two big leaves (see Figure 2), one representing the seasonal vegetation (grasses), and one representing the perennial vegetation (trees). Currently, the VOM does not explicitly consider leaf area dynamics, but the photosynthesis of both the seasonal and perennial vegetation was modelled with a simplified canopy-gas exchange model of von Caemmerer (2000) for C$_3$-plants. This was done by Schymanski et al. (2009) for a greater generality of the VOM,

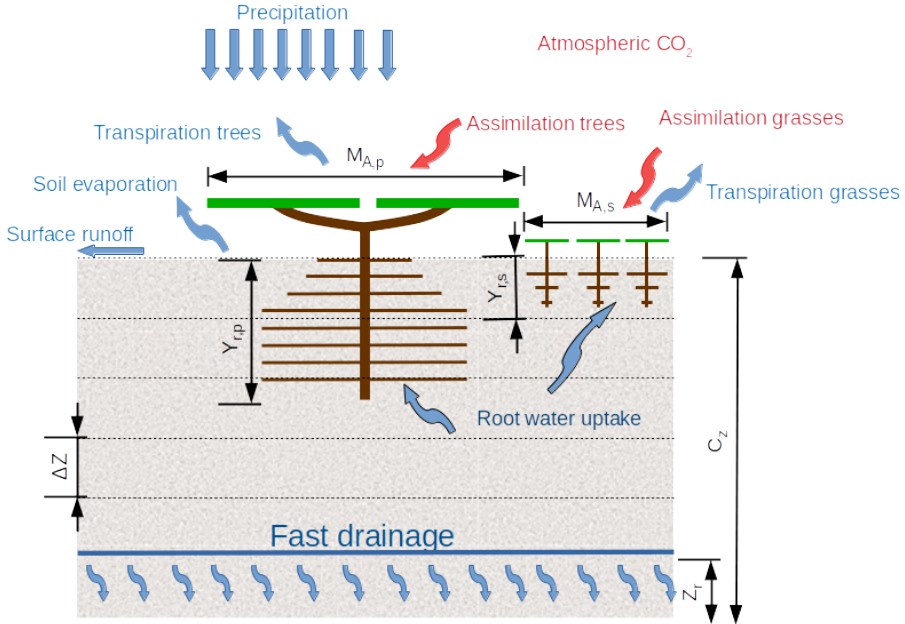

**Figure 2.** Schematization of the Vegetation Optimality Model as two big leaves, with $M_{A,p}$ and $M_{A,s}$ the fractional cover of perennial trees and seasonal grasses respectively, $y_{r,p}$ and $y_{r,s}$ the rooting depths of the perennial trees and seasonal grasses respectively, $\Delta Z$ the soil layer thickness, $C_Z$ the total soil depth, and $Z_r$ the drainage depth.

even though it may not correctly represent photosynthesis of the $C_4$-grasses at the site. The model computes $CO_2$-uptake as a

function of irradiance, atmospheric $CO_2$-concentrations, photosynthetic capacity and stomatal conductance:

$$A_g = \frac{1}{8}\left(4C_aG_s + 8\Gamma_*G_s + \left((J_e - 4R_l - 4G_s(C_a - 2\Gamma_*))^2 + 16G_s(8C_aG_s + J_e + 8R_l)\Gamma_*\right)^{\frac{1}{2}}\right) \qquad (2)$$

with $J_e$ the electron transport rate (mol m$^{-2}$ s$^{-1}$), $G_s$ stomatal conductance (mol m$^{-2}$ s$^{-1}$), $R_l$ leaf respiration (mol m$^{-2}$ s$^{-1}$), $C_a$ the mole fraction of $CO_2$ in the air and $\Gamma_*$ the $CO_2$ compensation point (mol $CO_2$ mol$^{-1}$ air). The electron transport rate $J_e$ was calculated as:

$$J_e = \left(1.0 - e^{\frac{0.3I_a}{J_{max}}}\right) \cdot J_{max} \cdot M_a \qquad (3)$$

with $I_a$ the irradiance (mol m$^{-2}$ s$^{-1}$), $J_{max}$ the electron transport capacity (mol m$^{-2}$ s$^{-1}$) and $M_a$ the projected cover of vegetation (dimensionless fraction). The leaf respiration $R_l$, as used in Equation 2, is defined as:

$$R_l = \frac{M_a \cdot c_{Rl} \cdot J_{max} \cdot (C_a - \Gamma_*)}{8 \cdot (C_a + 2 \cdot \Gamma_*)} \qquad (4)$$

with $c_{Rl}$ a constant set to 0.07 (dimensionless), as defined by Schymanski et al. (2007) and based on the results of Givnish (1988), who found that leaf respiration equals 7% of the maximum photosynthetic capacity for a range of different species. The electron transport capacity $J_{max}$ in Equations 3 and 4 is determined in the following way:

$$J_{max} = \frac{J_{max,25}\left(h_a\left(e^{-\frac{h_d(T_{opt}-298.0)}{273 T_{opt} R + 25.0}}-1.0\right)+h_d\right)e^{\frac{h_a(T_a-25.0)(273.0 T_{opt} R + T_{opt}-273.0)}{(T_a + 273.0 T_{opt} R)(273.0 T_{opt} R + 25.0)}}}{h_a\left(e^{\frac{h_d(T_a - T_{opt} + 273.0)}{T_a + 273.0 T_{opt} R}}-1.0\right)+h_d} \tag{5}$$

with $h_a$ the rate of exponential increase of the function below the $J_{max,25}$ and $h_d$ the rate of exponential decrease of the function above $J_{max,25}$, set to 43.79 and 200 kJ mol$^{-1}$, respectively (values taken for *Eucalyptus pauciflora*, see Schymanski et al., 2007; Medlyn et al., 2002). Furthermore, $J_{max,25}$ is the electron transport capacity at 25 $^o$C (mol m$^{-2}$ s$^{-1}$) and $T_{opt}$ the optimal temperature (K), set to the mean monthly daytime temperature at the site (305 K, Schymanski et al., 2007). In the equations above, the electron transport capacity at 25 $^o$C $J_{max,25}$, the projected foliage cover $M_a$ and stomatal conductance $G_s$ are optimized dynamically in a way to maximize the overall NCP of the vegetation over the entire simulation period. Optimization is possible due to the carbon costs associated with each of these variables: photosynthetic capacity is linked to maintenance respiration, projected cover is linked to foliage turnover and maintenance costs, while stomatal conductance is linked to transpiration (depending on the atmospheric vapour pressure deficit) and hence root water uptake costs and limitations. Root water uptake ($Q_{r,i}$, m/s), is modelled following an electrical circuit analogy, where the water potential difference between the plant and each soil layer drives the flow:

$$Q_{r,i} = S_{A,r}\frac{h_{r,i} - h_i}{\Omega_r + \Omega_{s,i}} \tag{6}$$

with $S_{A,r}$ the root surface area (m$^2$ m$^{-2}$), $h_{r,i}$ the hydraulic head in the roots (m), $h_i$ the hydraulic head in the soil (m), $\Omega_r$ the radial root resistivity (s) and $\Omega_{s,i}$ the soil resistivity (s), with subscript $i$ denoting the specific soil layer. The root surface area, $S_{A,r}$, is optimized in a way to satisfy the canopy water demand with the minimum possible total root surface area.

### 2.2.3 Carbon cost functions and Net Carbon Profit

As mentioned above, different carbon cost functions are used to quantify the maintenance costs for different plant organs. The carbon cost related to foliage maintenance ($R_f$) is based on a linear relation between the total leaf area and a constant leaf turnover cost factor:

$$R_f = L_{AIc} \cdot c_{tc} \cdot M_{A,p} \tag{7}$$

where $L_{AIc}$ is the clumped leaf area index (LAI of vegetated area, set to a constant 2.5 based on Schymanski et al., 2007), $c_{tc}$ is the leaf turnover cost factor (set to 0.22 $\mu$mol$^{-1}$ s$^{-1}$ m$^{-2}$, after an analysis of the Glopnet dataset (Wright et al., 2004) by Schymanski et al. (2007)) and $M_{A,p}$ is the perennial vegetation cover fraction.

The costs for root maintenance ($R_r$) were defined as:

$$R_r = c_{Rr} \cdot \left( \frac{r_r}{2} \cdot S_{A,r} \right) \tag{8}$$

where $c_{Rr}$ is the respiration rate per fine root volume (0.0017 mol s$^{-1}$ m$^{-3}$), $r_r$ the root radius (set to 0.3*10$^{-3}$ m), which were both derived by Schymanski et al. (2008b) from experimental data on citrus plants. Here, we present a sensitivity analysis of these parameters in Supplement S5. $S_{A,r}$ represents the root surface area per unit ground area (m$^2$m$^{-2}$).

Water transport costs are assumed to depend on the size of the transport system, from fine roots to the leaves. The canopy height is not modelled in the VOM, and the transport costs ($R_v$) are therefore just a function of rooting depth and vegetated cover:

$$R_v = c_{rv} \cdot M_A \cdot y_r \tag{9}$$

where $c_{rv}$ is the cost factor for water transport (mol m$^{-3}$ s$^{-1}$), $M_A$ the fraction of vegetation cover (−), and $y_r$ the rooting depth (m). The cost factor $c_{rv}$ was set to 1.0 $\mu$mol m$^{-3}$ s$^{-1}$ by Schymanski et al. (2015) after a sensitivity analysis for Howard Springs, which is also adopted here.

Based on the carbon cost functions and the assimilated carbon by photosynthesis ($A_g$) we can calculate the Net Carbon Profit:

$$NCP = \int \left( A_g(t) - R_f(t) - R_r(t) - R_v(t) \right) dt \tag{10}$$

with $t$ representing the time step.

### 2.2.4 Long-term optimization

The rooting depths of the perennial trees and the seasonal grasses ($y_{r,p}$ and $y_{r,s}$) as well as the foliage projected cover of the perennial vegetation ($M_{A,p}$) are derived by optimizing these properties for the long-term, assuming that these do not vary significantly during the simulation period (20-30 years). Similarly, water use strategies of both the perennial and the seasonal vegetation components are assumed to be a result of long-term natural selection for a given site, and are also optimized in order to maximize the NCP. To do so, the water use strategy was expressed as a functional relation between the marginal water cost of assimilation (Cowan and Farquhar, 1977), represented by $\lambda_p$ and $\lambda_s$ (mol mol$^{-1}$ m$^{-1}$) for perennial and seasonal vegetation respectively, and the sum of water suction heads ($h_i$) in all soil layers within the root zone:

$$\lambda_s = c_{\lambda f,s} \left( \sum_{i=1}^{i_{r,s}} h_i \right)^{c_{\lambda e,s}} \tag{11}$$

$$\lambda_p = c_{\lambda f,p} \left( \sum_{i=1}^{i_{r,p}} h_i \right)^{c_{\lambda e,p}} \tag{12}$$

where $c_{\lambda f,s}$ (mol mol$^{-1}$ m$^{-1}$), $c_{\lambda e,s}$ (-), $c_{\lambda f,p}$ (mol mol$^{-1}$ m$^{-1}$) and $c_{\lambda e,p}$ (-) are the optimized parameters, while $i_{r,p}$ and $i_{r,s}$ represent the number of soil layers reached by perennial and seasonal roots, respectively. Note that Cowan and Farquhar (1977) proposed that $\lambda$ should decline with declining soil water content, whereas Schymanski et al. (2009) argued that plants more likely sense the soil suction head than the total available water. Equations 11 and 12 formulated $\lambda$ as an explicit but flexible function of the average suction head in the root zone, where the shape of the function (determined by the two optimized parameters) represents a specific water use strategy.

After the establishement of the optimized water use parameters in Eqs. 11 and 12, the values of $\lambda_p$ and $\lambda_s$ are calculated for each day separately and then used to simulate the diurnal variation in stomatal conductance using Cowan-Farquhar optimality (Cowan and Farquhar, 1977; Schymanski et al., 2008a). The values of $c_{\lambda f,s}$, $c_{\lambda e,s}$, $c_{\lambda f,p}$ and $c_{\lambda e,p}$ express how quickly plants reduce water use as soil water suction increases during dry periods. The parameters ($c_{\lambda f,s}$, $c_{\lambda e,s}$, $c_{\lambda f,p}$ and $c_{\lambda e,p}$) are optimized and constant in the long-term, along with $y_r$ and $M_{A,p}$, to maximize the total NCP over the entire simulation period.

### 2.2.5  Short-term optimization

Some vegetation properties, such as seasonal vegetation cover ($M_{A,s}$), the electron transport capacities at 25 $^o$C for the seasonal and perennial vegetation ($J_{max25,p}$ and $J_{max25,p}$) and root surface area distributions of the seasonal and perennial vegetation ($S_{Adr,i,s}$ and $S_{Adr,i,p}$), are allowed to vary on a daily basis to reflect their dynamic nature. Here, the root surface area distributions are optimized day by day in a way to satisfy the canopy water demand. In a first step, this is done by determining a coefficient of change $k_r$, defined as:

$$k_r = \frac{0.95 M_{qx} - M_{q,min}}{0.05 \cdot M_{qx}} \tag{13}$$

with $M_{qx}$ the maximum tissue water content (kg m$^{-2}$) and $M_{q,min}$ the minimum daily tissue water content (kg m$^{-2}$). The minimum daily tissue water content is not allowed to be less than 0.9*$M_{qx}$ (i.e. stomata close when it approaches this value) and can not exceed the maximum tissue water content $M_{qx}$. The coefficient of change ($k_r$) ranges between 1 if maximum tissue water depletion was reached during the day (if $M_{q,min} = 0.9 * M_{qx}$) and -1 if tissue water was not depleted at all ($M_{q,min} = M_{qx}$). After a value of $k_r$ is calculated at the end of a day, the relative effectiveness of the roots in the different layers $k_{reff,i}$ is evaluated:

$$k_{r,eff,i} = \frac{0.5 \cdot \frac{Q_{r,daily,i}}{S_{A,r,i}}}{max \left( \frac{Q_{r,daily,1}}{S_{A,r,1}}, ..., \frac{Q_{r,daily,n}}{S_{A,r,n}} \right)} \tag{14}$$

with $Q_{r,daily,i}$ the daily root water uptake in layer $i$ (m/s), $S_{A,r,i}$ the root surface area in layer $i$ (m$^2$ m$^{-2}$) and $n$ the number of layers. Eventually, the new root surface areas per layer are determined based on the factors $k_r$ and $k_{r,eff}$ and the maximum growth per day ($G_r, max$, set to a value of 0.1 m$^2$ m$^{-3}$, see Schymanski et al., 2008b):

$$\Delta S_{A,r,i} = G_{r,max} \cdot k_r \cdot k_{r,eff,i} \tag{15}$$

The other vegetation properties are optimized from day to day in a way to maximize the daily NCP. This is done by using three different values for each of these vegetation properties, the actual value and a specific increment above and below this value every day, and at the end of the simulated day the combination of values that would have achieved the maximum NCP on the present day is selected for the next day. These vegetation parameters always vary on a daily basis, even though the time step 235 of the VOM is usually hourly or sub-hourly. Only the stomatal conductances, as determined by Cowan-Farquhar optimality, are varied over an hourly time step.

### 2.2.6 Model optimization

The VOM uses the Shuffled Complex Evolution algorithm (SCE, Duan et al., 1994) to optimize the vegetation properties listed in Table 3 for maximum NCP over the entire simulation period (37 years for the VOM-v0.5, from 1-1-1980 until 31-12-2017). 240 The SCE-algorithm uses first an initial random seed, subdivides the parameter sets into complexes and performs a combination of local optimization within each complex and mixing between complexes to converge to a global optimum. Here, we set the initial number of complexes to 10. The VOM uses a variable time step, where the target time step length of 1 hour is reduced if any state variable in the model changes by more than 10% per time step.

### 2.2.7 Meteorological data

A relatively long timeseries of meteorological inputs is required to run and optimize the VOM. The necessary meteorological data includes daily time series of maximum and minimum temperatures, shortwave radiation, precipitation, vapour pressure and atmospheric pressure, which were taken from the Australian SILO Data Drill (Jeffrey et al., 2001). In addition, the VOM requires information about atmospheric $CO_2$-levels, which were provided as a constant in the VOM-AoB2015 version, whereas for the VOM-v0.5 used here, we used the Mauna Loa $CO_2$-records (Keeling et al., 2005), see also Sect. 2.2.9. Eventually, these 250 daily time series are converted in the VOM to hourly time series, with a diurnal variation that was imposed for the global radiation, temperature and vapour pressure deficit, as described in detail in Appendix A of Schymanski et al. (2009).

Observed atmospheric $CO_2$-levels at the flux tower were not used due to the required length of the timeseries for the VOM (20-30 years). The measured meteorological variables at the flux tower sites were only used to verify the SILO meteorological data, which revealed only minor differences in the resulting fluxes of the VOM when the SILO-data was replaced for the days 255 that flux tower observations were available (max. 6%, see Figure S4.3 in Supplement S4). See also Supplement S3, Figure S3.1 for the time series of meteorological data.

**Table 2.** Vertical profile of soil characteristics at Howard Springs, based on data from the Soil and Landscape Grid of Australia (Viscarra Rossel et al., 2014a,b,c), in addition to field measurements of J. Beringer and L. B. Hutley. Here, $\theta_r$ refers to the residual moisture content, $\theta_s$ the saturated water content, $\alpha$ and $n$ the Van Genuchten soil parameters (van Genuchten, 1980) and $K_{sat}$ the saturated hydraulic conductivity.

| Howard Springs | Soil type | $\theta_r$ (-) | $\theta_s$ (-) | $\alpha$ (1/m) | $n$ (-) | $K_{sat}$ (m/s) |
|---|---|---|---|---|---|---|
| 0.00-0.20m | Sandy Loam | 0.065 | 0.41 | 7.5 | 1.89 | $1.228 * 10^{-5}$ |
| 0.20-0.40m | Sandy Loam | 0.065 | 0.41 | 7.5 | 1.89 | $1.228 * 10^{-5}$ |
| 0.40-0.60m | Sandy Clay Loam | 0.1 | 0.39 | 5.9 | 1.48 | $3.639 * 10^{-6}$ |
| 0.60-bedrock | Sandy Clay Loam | 0.1 | 0.39 | 5.9 | 1.48 | $3.639 * 10^{-6}$ |

### 2.2.8 Model evaluation data

At Howard Springs, a flux tower that is part of the regional FLUXNET network OzFlux (Beringer et al., 2016), provides time series of net ecosystem exchange (NEE) of carbon dioxide and latent heat flux (LE) for model evaluation. The Dingo-algorithm (Beringer et al., 2017) was applied to the data for a gap-filled estimation of gross primary productivity (GPP) and latent heat flux (LE). LE was converted to evapo-transpiration (ET), defined here as the sum of all evaporation and transpiration processes, even though these processes are different in nature (Savenije, 2004). Eventually, the gap-filled observations of GPP and ET were compared with the modelled fluxes. The VOM-AoB2015 was originally run until the end of 2005, and for this reason, the modelled fluxes were evaluated for the overlapping period between model and flux tower observations from 07-08-2001 until 31-12-2005. For consistency, the VOM-v0.5 runs were evaluated for the same time period.

To evaluate the foliage projected cover (FPC) dynamics of seasonal and perennial vegetation predicted by the VOM, defined as the sum of $M_{A,p}$ and $M_{A,s}$, we used satellite-derived monthly fractions of Photosynthetically Active Radiation absorbed by vegetation (fPAR) from Donohue et al. (2008, 2013), which were converted into estimates of FPC. The maximum possible value of fPAR was defined as $0.95$ by Donohue et al. (2008) and relates to maximum projective cover (i.e. FPC = 1.0). The linear relation of FPC with fPAR-data (Asrar et al., 1984; Lu, 2003) allowed for the calculation of FPC by dividing the fPAR-values by the maximum value of $0.95$.

### 2.2.9 Systematic analysis of modifications to the VOM set-up

To assess the effect of the modifications to the VOM set-up from version VOM-AoB2015 to VOM-v05, each individual change was added to the reference set-up of the VOM-AoB2015 (see also Supplement S1). We define here 12 model cases, including the 9 changes to the VOM-AoB2015, the reproduction of the VOM-AoB2015, the re-optimization of the VOM-AoB2015 and the final VOM-v0.5 (see also Table 4):

  – 1. Reproduction of the results of Schymanski et al. (2015)

The model code of the VOM-v0.5 was run with the same vegetation parameters and input data as the VOM-AoB2015. The model was run from 1-1-1976 until 31-12-2005. This was done in order to check the reproducibility of the results of Schymanski et al. (2015).

   – 2. Re-run SCE

The VOM was re-optimized with the same settings and input data as the VOM-AoB2015, and run with the same model period from 1-1-1976 until 31-12-2005. Also here, the specific goal was to reproduce the results of Schymanski et al. (2015), as the optimization algorithm should converge to the same solutions.

   – 3. Variable $CO_2$-levels

Atmospheric $CO_2$-levels were originally assumed constant in the VOM-AoB2015 with $CO_2$-concentrations of 350 ppm, but in the VOM-v0.5, these were taken from the Mauna Loa $CO_2$-records (Keeling et al., 2005). Therefore, the VOM-AoB2015 was run with variable $CO_2$-levels, and optimized for the period 1-1-1976 until 31-12-2005. This was done in order to assess the sensitivity of the model to variable $CO_2$-levels.

   – 4. Reduced soil layer thickness

The soil layer thickness was set to 0.2 m, instead of the 0.5 m used in the VOM-AoB2015, after running a sensitivity analyses with the VOM-v0.5 (see Supplement S2). The VOM-AoB2015 was also run with 0.2 m now, and optimized for the period 1-1-1976 until 31-12-2005, in order to assess the influence of different soil layer thicknesses on the VOM-AoB2015 results.

   – 5. Variable atmospheric pressure

A new version of the meteorological data from the Australian SILO Data Drill (Jeffrey et al., 2001) provided time series of atmospheric pressure starting from 1-1-1980, whereas originally this had been fixed at a level of 1013 hPa for the VOM-AoB2015. The variable atmospheric pressure was included in the VOM-AoB2015 and the model was optimized for the period 1-1-1980 until 31-12-2005, due to the available time series from 1-1-1980. This was performed to assess the importance of precise atmospheric pressure data for the VOM simulations.

   – 6. Optimized grass rooting depth

The rooting depth of grasses was prescribed at 1.0 m in the VOM-AoB2015, which is roughly the position of a hard pan in the soil profile at Howard Springs. In the VOM-v0.5, grass rooting depth is optimized along with the tree rooting depth in order to let also the grass rooting depths adapt to local conditions. To assess the effect of an optimized grass rooting depth separately, we also optimized it in the VOM-AoB2015 simulations for the period 1-1-1976 until 31-12-2005.

   – 7. Updated meteorological data

A new version of the meteorological data from the Australian SILO Data Drill (Jeffrey et al., 2001) was used in the VOM-v0.5, starting from 1-1-1980. Therefore, the VOM-AoB2015 was also optimized with the new meteorological data,

for the period 1-1-1980 until 31-12-2005. The time series of daily maximum and minimum temperatures, shortwave radiation, precipitation, vapour pressure and atmospheric pressure were all updated, but the $CO_2$-concentrations kept fixed at 350 ppm. In this way, it can be assessed to what extent the different data versions lead to different model results.

– 8. Updated and extended meteorological data

The new version of the meteorological data from the Australian SILO Data Drill (Jeffrey et al., 2001) includes more recent years, which were included in the VOM-v0.5 simulations. To find out in how far the inclusion of more recent meteorological forcing alone affected the results, we also re-ran the VOM-AoB2015 with the extended meteorological forcing but constant atmospheric $CO_2$-concentration of 350 ppm. Therefore, the model period of the VOM-AoB2015 was extended and the model was optimized from 1-1-1980 until 31-12-2017.

– 9. Modified hydrology

In the VOM-AoB2015, the average slope of the seepage face $\gamma_0$ and hydrological length scale $\Lambda_s$ were adopted from Reggiani et al. (2000) and set to 0.033 rad and 10 m, respectively, in absence of more detailed knowledge about these parameters. At the same time, the drainage level $z_r$ and total soil thickness $c_z$ were set to 10 m and 15 m, respectively, based on the local topography around the flux tower site (Schymanski et al., 2008b). This hydrological schematization resulted in groundwater tables around 5 m below the surface.

The hydrological parameters for the VOM-v0.5 were set in a way to resemble freely draining conditions, i.e. avoiding a significant influence of groundwater in Figure 2, required for a systematic comparison with other model applications in the accompanying paper of Nijzink et al. (2021), with a total soil thickness $c_z$ of 30 m, a fast drainage parameterization with a drainage level $z_r$ of 5 m (i.e. 25 m below the surface), a length scale for seepage outflow $\Lambda_s$ set to 2 m and a channel slope $\gamma_0$ set to 0.02 rad.

Therefore, the VOM-AoB2015 was optimized with the new hydrological schematization of the VOM-v0.5 for the period 1-1-1976 until 31-12-2005. In this way, the effect of the hydrological settings on the model results can be assessed.

– 10. Modified soil properties

The soils were assumed vertically homogeneous in the VOM-AoB2015, but were parameterized in the VOM-v0.5 based on field measurements of sand, clay and silt content provided by L. B. Hutley and J. Beringer in the top 10 cm, and the Soil and Landscape Grid of Australia (Viscarra Rossel et al., 2014a,b,c) for the deeper layers. The soils were classified into one of the soil textural groups of Carsel and Parrish (1988) based on the fractions of sand, silt and clay. Eventually, the parameters for the soil water retention model of van Genuchten (1980) and the hydraulic conductivity were taken from the accompanying tables[7] from Carsel and Parrish (1988). See also Table 2 for the soil parameterization in the VOM-v0.5. As a result, the soil profile at Howard Springs is now assumed to consist of sandy loam in the top 0.4 m and sandy clay loam below, whereas VOM-AoB2015 used a soil of sandy loam in the entire soil profile.

---

[7]see also https://vom.readthedocs.io/en/latest/soildata.html

**Table 3.** Vegetation properties in the Vegetation Optimality Model optimized for maximizing the NCP.

| Parameter | Description | Initial range | Timescale | Unit |
|---|---|---|---|---|
| $c_{\lambda f,p}$ | multiplicative water use parameter perennial vegetation | 0.0 - 10000.0 | Long-term | mol mol$^{-1}$ m$^{-1}$ |
| $c_{\lambda e,p}$ | exponential water use parameter perennial vegetation | -3.0 - 0.0 | Long-term | - |
| $c_{\lambda f,s}$ | multiplicative water use parameter seasonal vegetation | 0.0 - 10000.0 | Long-term | mol mol$^{-1}$ m$^{-1}$ |
| $c_{\lambda e,s}$ | exponential water use parameter seasonal vegetation | -3.0 - 0.0 | Long-term | - |
| $M_{A,p}$ | fractional cover perennial vegetation | 0 - 1 | Long-term | - |
| $y_{r,p}$ | rooting depth perennial vegetation | 1.0 - 9.0 | Long-term | m |
| $y_{r,s}$ | rooting depth seasonal vegetation | 0.05 - 2 | Long-term | m |
| $M_{A,s}$ | fractional cover seasonal vegetation | 0.00 - (1.0-$M_{A,p}$) | Daily | - |
| $J_{max25,p}$ | electron transport capacity perennial vegetation | - | Daily | mol s$^{-1}$ m$^{-2}$ |
| $J_{max25,s}$ | electron transport capacity annual vegetation | - | Daily | mol s$^{-1}$ m$^{-2}$ |
| $G_{s,p}$ | stomatal conductance perennial vegetation | - | Daily | mol s$^{-1}$ m$^{-2}$ |
| $G_{s,s}$ | stomatal conductance seasonal vegetation | - | Daily | mol s$^{-1}$ m$^{-2}$ |
| $S_{Adr,i,p}$ | root surface area distribution of perennial vegetation | - | Daily | m$^2$ m$^{-3}$ |
| $S_{Adr,i,s}$ | root surface area distribution of annual vegetation | - | Daily | m$^2$ m$^{-3}$ |

The VOM-AoB2015 was optimized here with the a modified soil profile, using the soil discretization of the VOM-AoB2015 of 0.5 m, with a sandy loam structure in the top layer and sandy clay loam below. This was done for the period 1-1-1976 until 31-12-2005, in order to assess the changes due to the different soils.

– 11. Modified soil properties and hydrology

The modified soils and hydrology, as described above, will strongly interact. Free draining conditions are expected to reduce soil water storage, while finer soil texture is expected to increase water storage. In order to better understand in how far these changes compensated for each other, they were both implemented together in the VOM-AoB2015, while keeping everything else unmodified.

– 12. VOM-v0.5

Eventually, all changes were applied to the VOM-AoB2015, resulting in the new VOM-v0.5 simulations, as presented in the accompanying paper by Nijzink et al. (2021).

**Table 4.** Modifications to the VOM-AoB2015.

| Case | Base model | Modification | Model period |
|------|-----------|--------------|--------------|
| 1. Reproduction VOM-AoB2015 | VOM-AoB2015 | None | 1-1-1976 – 31-12-2005 |
| 2. Re-run SCE VOM-AoB2015 | VOM-AoB2015 | None | 1-1-1976 – 31-12-2005 |
| 3. Variable $CO_2$-levels | VOM-AoB2015 | Constant $CO_2$ to variable | 1-1-1976 – 31-12-2005 |
| 4. Reduced soil layer thickness | VOM-AoB2015 | Soil layers from 0.5 m to 0.2 m | 1-1-1976 – 31-12-2005 |
| 5. Variable atmospheric pressure | VOM-AoB2015 | Constant pressure of 1013 hPa to variable | 1-1-1980 – 31-12-2005 |
| 6. Optimized grass rooting depth | VOM-AoB2015 | Grass roots from 1 m to optimized | 1-1-1976 – 31-12-2005 |
| 7. Updated meteorological data | VOM-AoB2015 | New SILO-data (Jeffrey et al., 2001) | 1-1-1980 – 31-12-2005 |
| 8. Updated/extended meteorological data | VOM-AoB2015 | New/extended SILO-data (Jeffrey et al., 2001) | 1-1-1980 – 31-12-2017 |
| 9. Modified hydrology | VOM-AoB2015 | New hydrological schematization | 1-1-1976 – 31-12-2005 |
| 10. Modified soils | VOM-AoB2015 | Update soil profiles | 1-1-1976 – 31-12-2005 |
| 11. Modified soils and hydrology | VOM-AoB2015 | New hydrological schematization and soils | 1-1-1976 – 31-12-2005 |
| 12. VOM-v0.5 | VOM-v0.5 | All modifications included | 1-1-1980 – 31-12-2017 |

## 3   Results and Discussion

### 3.1   Effects of modifications to the VOM

To compare previous simulations using the VOM-AoB2015 with the VOM-v0.5 set-up that includes the modifications as outlined in Sect. 2.2.9, each modification was applied to the previous setup in a one-step-at-a-time approach to quantify the in-
fluence of each change in isolation. The resulting simulations were compared with those presented in Schymanski et al. (2015) for the site Howard Springs. In general, sensitivities varied between +20% and -25% in total GPP and ET after optimizing the VOM with the new changes, and are summarized in Figure 3. Without optimizing the VOM, but still including the modifications, the sensitivities in GPP and ET were even smaller (see Supplement S1, Figure S1.51), except for a strong increase in soil evaporation after changing the soils. See also Supplement S1 for detailed time series.

As expected, re-running the VOM-AoB2015 (Case 1 in Table 4) with the originally optimized parameters resulted in neg-ligible differences. Re-running the optimization (Case 2) did result in slightly different results (12.6% higher projective cover for the perennials), but none of the simulated fluxes changed by more than 10% (Fig. 3). Similar to the fluxes, the changes in vegetation parameters for reproducing VOM-AoB2015 and re-running the optimization algorithm remained small (Figure 4).

In contrast, changing the fixed atmospheric $CO_2$-levels (350 ppm) in the VOM-AoB2015 to variable atmospheric $CO_2$-
levels (Case 3) had a relatively large influence on perennial vegetation, yielding values of GPP for perennial vegetation that were up to 21.0% higher (Figure 3d). Note that the $CO_2$-levels of the Mauna Loa records have a mean of 369 ppm and a maximum of 410 ppm during the modelling period, i.e. mostly higher than the 350 ppm prescribed in the VOM-AoB2015. See Figure S3.1f in Supplement S3 for more details about the $CO_2$-levels used here. Interestingly, the implementation of variable

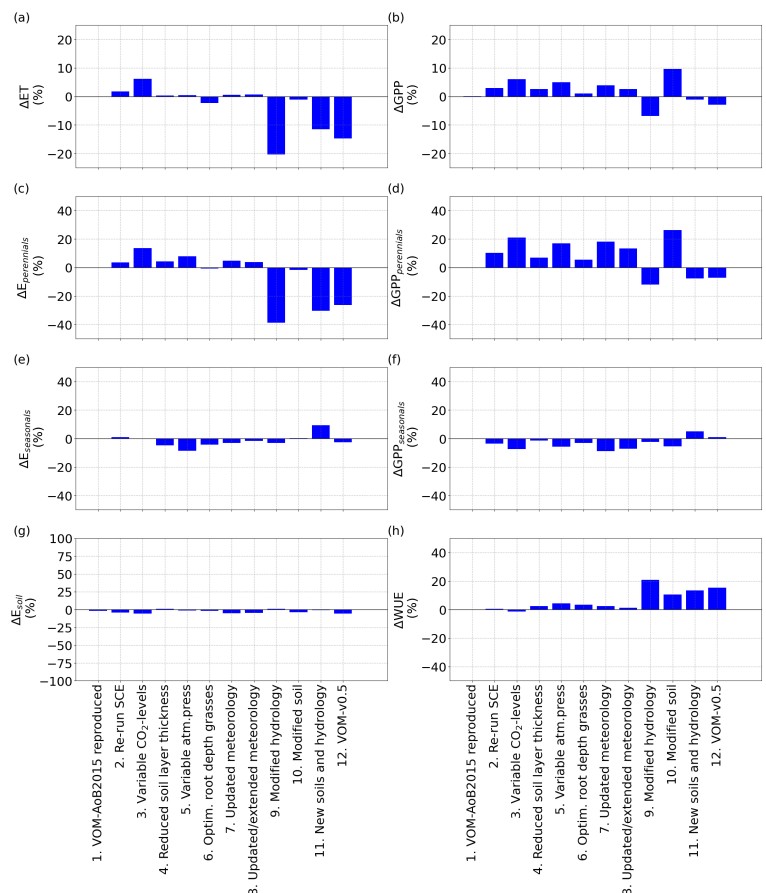

**Figure 3.** Relative changes in the mean annual values of the fluxes for the different (incremental) changes, as described in sect. 2.2.9, in comparison with the VOM-AoB2015, for a) ET, b) GPP, c) transpiration perennials (trees), d) GPP perennials (trees), e) transpiration seasonals (grasses), f) GPP seasonals (grasses), g) soil evaporation and h) combined water use efficiency (WUE) of seasonal and perennial vegetation.

CO$_2$-concentrations led to a large increase in $c_{\lambda e,p}$, i.e. one of the water use strategy parameters of the perennials (Fig. 4a).

Another effect was a larger perennial vegetation cover (Figure 4e), while the seasonal cover was on average reduced, which relates to the generally elevated CO$_2$-levels and the long-term optimization of the perennial vegetation cover. Hence, perennial vegetation cover benefited, and the grass cover, optimized on a daily basis, reduced on average (Figure 4f).

Changing the vertical soil discretization of 0.5 m in the VOM-AoB2015 to a finer resolution of 0.2 m (Case 4) had a minor influence, with a change less than the variability due to re-running the optimization algorithm (i.e. 2.6 % in the resulting GPP

and 0.3 % in ET, Figure 3b, a). The reduced soil layer thickness mainly affected the water use parameters $c_{\lambda e,p}$ and $c_{\lambda e,s}$ (Figure 4a and b, respectively). At the same time, the root depths for the perennials increased (Figure 4g), which compensated for the change in water use, resulting in only minor changes in the fluxes (Figure 3).

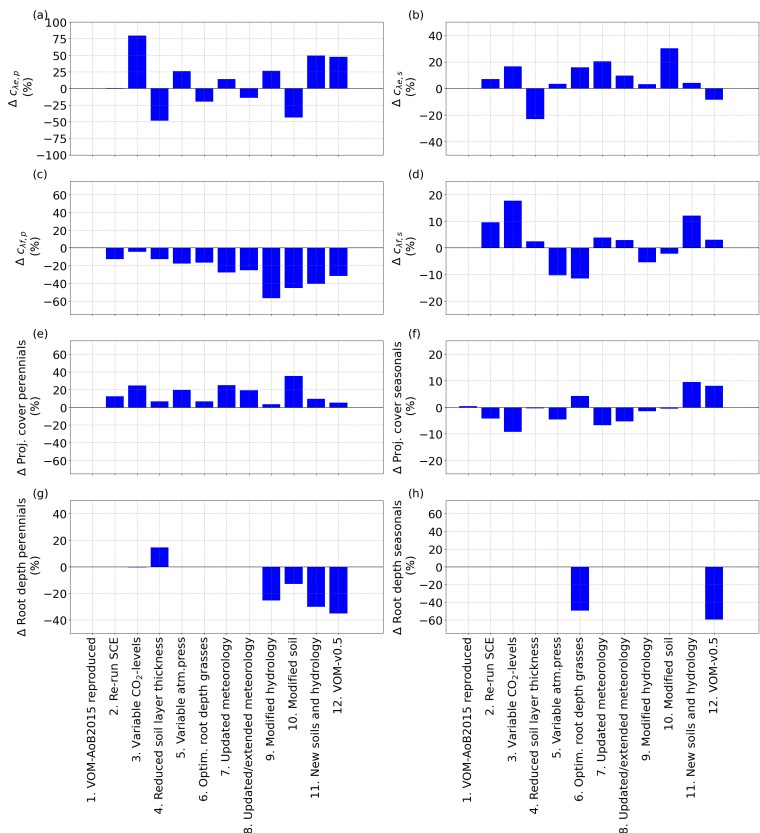

**Figure 4.** Relative changes in the vegetation properties for the different (incremental) changes, as described in sect. 2.2.9, in comparison with the VOM-AoB2015, for a) the water use parameter $c_{\lambda e,p}$ for perennial vegetation, b) the water use parameter $c_{\lambda e,s}$ for seasonal vegetation, c) the water use parameter $c_{\lambda f,p}$ for perennial vegetation, d) the water use parameter $c_{\lambda f,s}$ for seasonal vegetation, e) projected cover perennials (trees), f) mean projected cover seasonals (grasses), g) root depth perennials (trees) and h) root depth seasonals (grasses).

The variable atmospheric pressures (Case 5) only had a minor influence as well (5.0 % change in GPP and 0.4 % change in ET), which could also relate to re-running the optimization algorithm. It led to changes in the vegetation parameters as well,
but this stayed limited to a maximum of 25 % for $c_{\lambda e,p}$ (Figure 4a). However, this is stronger than the observed changes in the resulting fluxes of ET and GPP (Figure 3), which remained much smaller, related also to the non-linear relationship between $c_\lambda$, ET and GPP.

Similarly, when the grass rooting depths were optimized (Case 6) instead of the prescribed rooting depth of 1 m, simulated GPP and ET were changed by 1.0% and -2.3% respectively (Figure 3b and a). The optimization led to shallower grass roots
of 0.5 m (incurring lower carbon costs) and, therefore, to reductions in GPP and ET. This was accompanied by increased $c_{\lambda e,s}$ and reduced $c_{\lambda f,s}$ (Figure 4b and d), pointing at a more efficient water use strategy with less water transpired per assimilated $CO_2$ (Figure 3h).

The updated meteorological input data, for the runs until 31-12-2005 (Case 7) as well as the extended runs until 31-12-2017 (Case 8), hardly influenced the outcomes, with less than 10% relative change in the resulting fluxes (Figure 3a and b). However, a higher contribution of the perennial vegetation in the fluxes can be observed, related to an increase in perennial vegetation cover (24.8 %, Figure 4e) from 31.7 % to 39.6 %. This happened as well for re-running the SCE-algorithm, and the changes related to the updated meteorological input could be attributed to re-running the optimization algorithm as well. However, also the water use strategy parameters (4a-d) changed, with especially a strong change for $c_{\lambda f,p}$ with -27.8 % (Figure 4c), but the resulting total water use efficiency remained again similar (Figure 3h).

The implementation of free draining conditions (Case 9) had strong effects on the simulated fluxes, with lower values of both ET and GPP (-20.4% and -6.9% respectively, Figure 3a and b). However, here especially the simulated transpiration of perennial vegetation was reduced, whereas the transpiration by seasonal vegetation stayed relatively similar (Figure 3c and e). This is because in the original simulations, capillary rise from the water table was most important during the dry season, when seasonal vegetation is inactive, and a change in the water table due to free draining conditions affects therefore mostly the perennial and not so much the seasonal vegetation. The modified hydrology led to larger changes in the vegetation properties, with a particularly strong decrease in $c_{\lambda f,p}$ (Figure 4c), and to a lesser degree $c_{\lambda f,s}$ (Figure 4d). Hence, the modified hydrology led to a more efficient water use (Figure 3h). At the same time, the root depths for the perennial vegetation strongly reduced as well (Figure 4g).

Even stronger effects were found for the updated soil texture, which resulted in slightly reduced ET (-1.1%), but clearly increased GPP by 9.6% (Figure 3a and b). The increase in simulated GPP was largely due to increased GPP by perennial vegetation, which at the same time slightly decreased its transpiration. This coincided with a largely increased perennial vegetation cover and reduced rooting depth compared to the original simulations (vegetation cover increased from 0.31 to 0.43, while rooting depth declined from 4 m to 3.5 m). Overall, the perennial vegetation benefited from the greater carry-over of soil moisture from the wet season into the dry season (Supplement S1, Figure 1.40c) due to finer soil texture and associated increased water retention and reduced drainability. More water availability during the dry season allowed an increased perennial vegetation cover to be maintained. On the other hand, the finer soil texture resulted in higher suction heads during the wet season (Supplement S1, Figure 1.40d), making water uptake more "expensive" as more fine roots would be needed to achieve the same water uptake rates as in coarser textured soil. This resulted in a new optimum with enhanced perennial vegetation cover and, therefore, increased GPP (Supplement S1, Figure 1.37f), while perennial water use was only moderately increased in the dry season, and even slightly reduced during the wet season (Supplement S1, Figure 1.37b), leading to largely increased water use efficiency per ground area (Figure 4h).

Combining the new soils with the new hydrological settings (Case 11) still resulted in a reduction in ET by -11.5%, whereas their combined effect on GPP led to only a small reduction by -1.1% (Figure 3a and b). Here, the reduction of ET occured mainly during the dry season, and related to reductions in the perennial transpiration (Figure 5a-c), whereas the GPP stayed relatively similar (Figure 5e). At the same time, updating the soils and hydrology combined, resulted in a more moderate increase of the perennial vegetation cover (Figure 4e). These findings are in accordance with the isolated effects of the new soils and hydrology, where free drainage conditions resulted in a large reduction in ET and GPP, while finer soil texture resulted

in a small reduction in ET but large increase in GPP. Hence, the finer soil texture largely compensated the effect on GPP, but not on ET.

## 3.2 Comparing VOM-v0.5 and VOM-AoB2015: resulting differences and underlying mechanisms

Eventually, all the changes were incorporated in the VOM-AoB2015 (Case 1) resulting in the VOM-v0.5 (Case 12), as used in the accompanying paper of Nijzink et al. (2021). Previously, we identified the isolated effects of these changes, but here we explore the combined effects of the most important modifications.

The relative error for mean annual evapo-transpiration (ET) changed for the VOM-AoB2015 from an overestimation by 8.4% to an underestimation of -10.2% for the VOM-v0.5, whereas the relative error for the mean annual GPP changed from 17.8% to 14.7 % overestimation. The GPP showed especially differences during the transition from the wet to the dry season (Figure 6b). Both the VOM-v0.5 and VOM-AoB2015 overestimated the GPP in comparison with the flux tower observations, but the modifications reduced this overestimation for the VOM-v0.5, mainly caused by a reduction of GPP by the perennial trees (Figure 3d). However, the changes in ET were stronger and the water use efficiency strongly increased (Figure 3h).

The ensemble years in Figure 6 revealed that the evapo-transpiration (ET) was most strongly underestimated by the VOM during the dry season at Howard Springs. The observed groundwater tables (Figure 7a) ranged from 5-15 m depth seasonally, whereas the VOM was parameterized now to keep groundwater tables close to 25 m depth, required for the accompanying paper of Nijzink et al. (2021). Schymanski et al. (2015) originally assumed a much shallower drainage level at Howard Springs, which led to groundwater tables around 5 meters depth, and better correspondence with the observed fluxes (Fig. 6).

The simulated soil moisture in the top soil layer, as illustrated in Fig. 7b, remained similar to the soil moisture values of Schymanski et al. (2015). The higher vertical resolution in the new model runs (20 cm cf. 50 cm soil layers) resulted in stronger surface soil moisture spikes around rainfall events, which makes the red line appear generally more noisy than the green line in Fig. 7b. Observed soil moisture in the upper 5 cm showed similar patterns as the simulated soil moisture in the top soil layer. The modelled soil moisture was generally higher, as this represents the integrated soil moisture over 0.5 m (VOM-AoB2015) and 0.2 m (VOM-v0.5), whereas the observed soil moisture in the top 5 cm is expected to be lower due to soil evaporation and percolation. The total water storage in the root zone was higher during the dry season in the new model simulations compared to Schymanski et al. (2015) (Figure 7c), but lower during the wet season. The water retention curves (Figure 8) also show a clear shift, especially for the layers below 0.4 m, indicating extra storage.

However, the water suction heads again showed strong similarities between the current model runs and the results of Schymanski et al. (2015) (Figures 7d and e, respectively), but with differences reflecting the vertical resolutions of the soil domains and simulated rooting depths, and generally higher values for the VOM-v0.5 for the deeper layers. Hence, also the water storage in the root zone increased during the transition from the wet to the dry period (Figure 7c), but resistivity increased as well due to lower hydraulic conductivities. Changing the hydraulic conductivity in isolation, clearly leads to reductions in ET for the perennials (Supplement S1, Figure S1.41b) due to the increased resistivity. Nevertheless, this is in the final VOM-v0.5 compensated for by the newly optimized water use strategy parameters and a higher water use efficiency (Figure 3h). However,

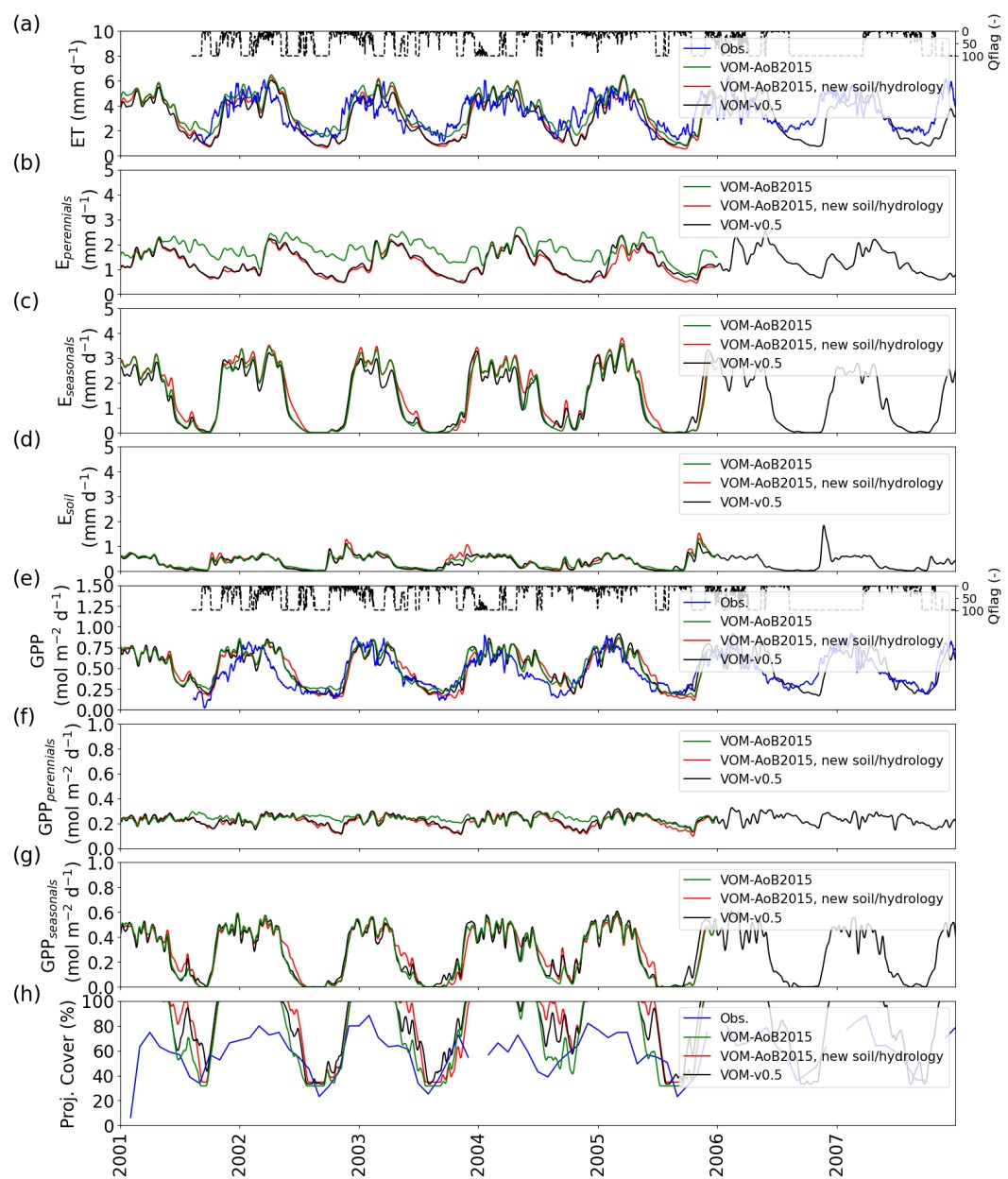

**Figure 5.** Comparison between the results of the VOM-AoB2015 in green, simulations using new soil and hydrological parameterization (red), and simulations using all changes in combination (black). a) ET, b) transpiration by perennials (trees), c) transpiration seasonals (grasses), d) soil evaporation, e) GPP, f) GPP perennials (trees), g) GPP seasonals (grasses), and h) projective cover. Time series in a)-g) were smoothed using a moving average of 7 days. The daily average quality flags of the flux tower observations are shown as a dashed line in Panel (e), with a value of 100 for a completely gap-filled day and 1 for gap-free observations.

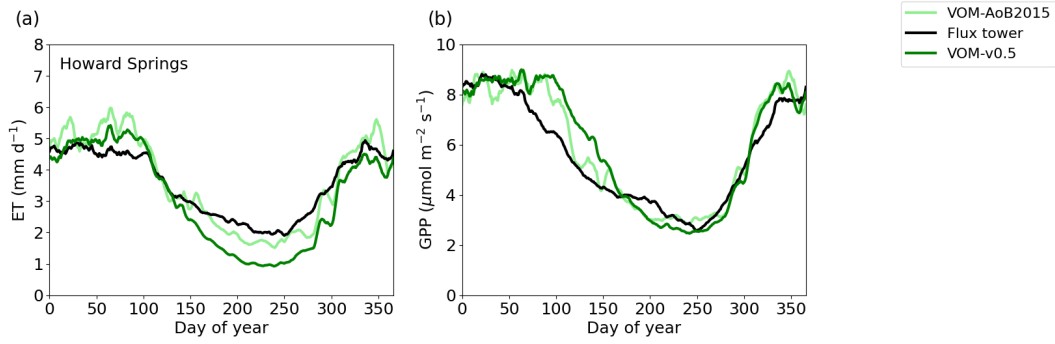

**Figure 6.** Ensemble years of evapo-transpiration (ET) and gross primary productivity (GPP) for the VOM-v0.5 (dark green), flux tower observations (black) and VOM-AoB2015 (light green), all smoothed by a 7-day moving average. The ensemble years are calculated for the overlapping time periods with the flux tower observations (7-8-2001 until 21-12-2016).

the new soil structure has the largest effect, as can be observed when running the VOM-v0.5 (i.e. with the new soils) with the water use strategy parameters of the VOM-AoB2015 in Figure S1.56b of Supplement S1.

## 4    Conclusions

As models, input data and parameterizations evolve, the effects of individual improvements are usually not systematically
evaluated. Here, we analyse the effects of changes to the Vegetation Optimality Model (VOM) between version VOM-AoB2015 and VOM-v0.5, the latter of which is the basis of a companion paper (Nijzink et al., 2021). Some of the modifications were done for improved realism, while others for better comparability with other models and general applicability across the different sites investigated.

The modifications consisted of updated and extended input data, the use of variable atmospheric $CO_2$-levels, modified soil
properties, modified drainage levels as well as the addition of grass rooting depths to the optimized vegetation properties. The changes were applied to the VOM in a one-step manner for the flux tower site Howard Springs, by applying each modification to the previous set-up of Schymanski et al. (2015) in isolation to evaluate its effect on the results, before combining all modifications and analyzing their effect in combination.

This analysis revealed that updated soil textures and a changed hydrological schematization had a strong influence on the
results (see Figures 3 and Figure 4). An underestimation of dry season ET at Howard Springs (Figure 6) was much more apparent when compared to the results of Schymanski et al. (2015), where the drainage parameterization maintained a water table depth much closer to the observed water table at this site. The effect of a much deeper groundwater table in the present simulations was partly buffered by a more fine-grained soil texture below 0.4 m (sandy clay loam instead of sandy loam), which resulted in an increase in field capacities (Figure 8) in the deeper soil layers compared to the simulations by Schymanski et al.
(2015). The use of variable atmospheric $CO_2$-levels also had a strong influence on the results, which is especially important as

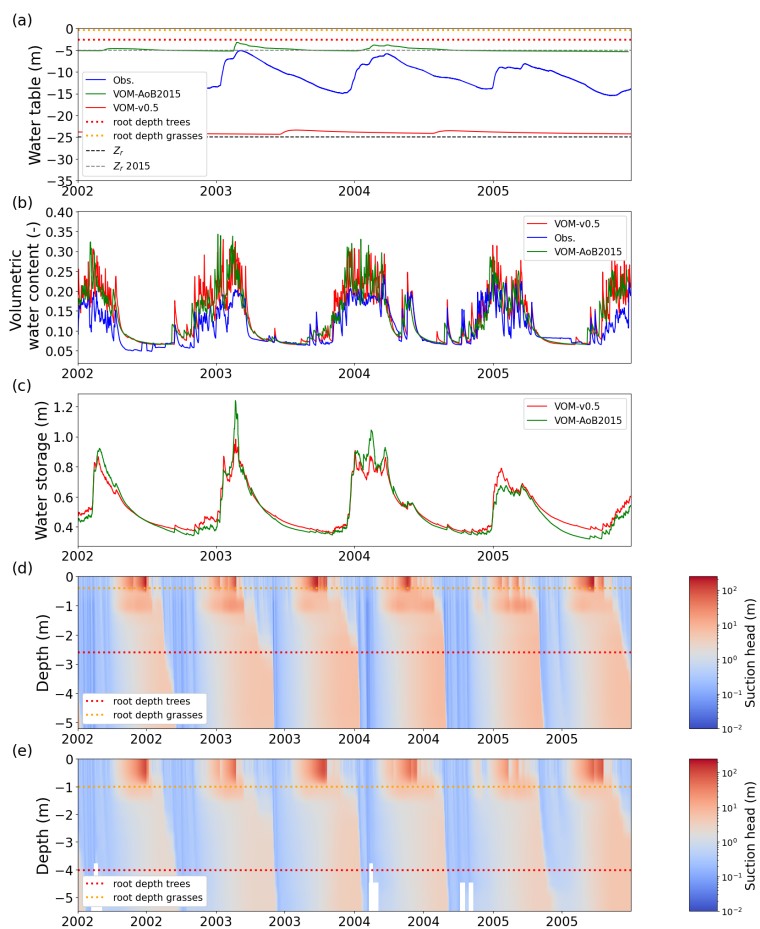

**Figure 7.** Simulated and observed hydrological state variables at Howard Springs. (a) groundwater depths, with dashed lines representing the prescribed bedrock depths, dotted lines the rooting depths (trees in red and grasses in orange), the VOM-v0.5 in red, the VOM-AoB2015 in green, and observations of three different bore holes in the vicinity of the study site in blue (Northern Territory Government, Australia, 2018); (b) the volumetric water content in the upper soil layer with the VOM results in red, the results of Schymanski et al. (2015) in green and measurement-based values at 5 cm depth at the flux tower sites in blue; (c) the total water storage in the root zone for the VOM-v0.5 (red, storage until 2.6 m) and the VOM-AoB2015 in green (storage until 4.0 m); (d) the suction heads for the VOM-v0.5; and (e) the suction heads of the VOM-AoB2015. Observed water tables in Panel a) were obtained from the Water data portal, (Northern Territory Government, Australia, 2018, accessed 8-3-2019).

the model time period has been extended in this study. This was mainly due to generally higher levels of atmospheric $CO_2$ in recent observations, compared to the constant values used by Schymanski et al. (2015).

Hence, our approach led to the insights that the neglect of a varying water table may have a strong effect on simulated surface fluxes. This is in line with other studies (e.g. York et al., 2002; Bierkens and van den Hurk, 2007; Maxwell et al., 2007), and shows that the common assumption of free draining conditions in modelling studies should be revised. Interestingly, the

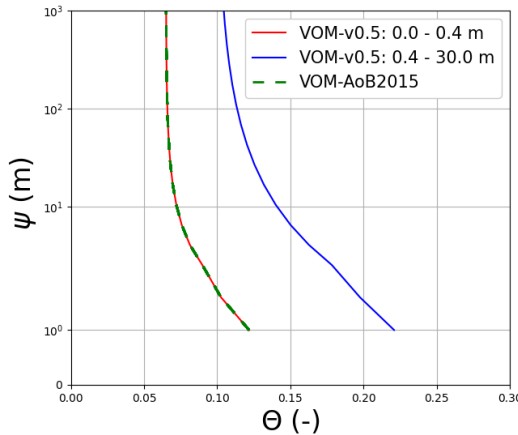

**Figure 8.** Water retention curves at Howard Springs for the VOM-v0.5 results (red: top two layers, blue: deeper layers), and the results of the VOM-AoB2015 in green. Note that multiple lines are shown for the VOM-v0.5 due to the different soil parameterizations per soil layer in the current model runs, whereas the VOM-AoB2015 used one soil parameterization for all soil layers. The upper soil layers have however the same soil parameters, leading to overlapping curves (red and dashed green).

deficiencies in TBMs related to water access and tree rooting depth as identified by Whitley et al. (2016) strongly relate to this, as root depths also depend on groundwater tables. At least, if a free draining assumption is necessary, due to a lack of better hydrological understanding of a given site, or for comparison with other model simulations, i.e. for the study in the accompanying paper of Nijzink et al. (2021), potential bias in simulation results has to be acknowledged.

In addition to this, we can more generally conclude that optimality-based modelling is able to provide robust modelling results. The sensitivities for the different changes remained rather limited and varied only between +20% and -25% in total GPP and ET after re-optimizing, but implementing the changes without re-optimizing the vegetation properties, resulted in even smaller changes. Therefore, we gained confidence that the VOM will also provide reliable and robust results for other study sites along the North Australian Tropical Transect, with similar boundary conditions. At the same time, the new boundary
conditions made the VOM comparable to the other TBMs of Whitley et al. (2016).

To conclude, with our analysis we identified more conceptual issues, e.g. the influence of the hydrological schematization, but also found confirmation that the optimality-based modelling approach provides a robust result. Our method is in line with other developments in terrestrial biosphere modelling, where benchmark testing is seen as a necessary step in the modelling process (Blyth et al., 2011; Abramowitz, 2012; Clark et al., 2021). Here, we provided an example on how to perform a
systematic benchmark test in a one-step-at-a-time approach, i.e. applying one change in isolation to the benchmark model. This analysis of modifications to a model setup and comparison against a benchmark dataset proved very helpful for identifying sensitivities of simulations to the different changes that might otherwise remain undiscovered due to compensating effects of the various modifications during model development. Our work also highlights the importance of open source code and data. The availability of the original VOM-AoB2015 code was a pre-requisite of conducting this analysis. The long-term storage

of the new VOM-v0.5, all input data, parameterization and data analysis workflows used in the present study will ensure that the effects of future modifications can again be compared step-by-step to the results presented here. This is a pre-requisite for maintaining generality of a model, as opposed to models that are highly customized for a particular site and hence unable to provide general insights.

*Code and data availability.* Model code is available on github (https://github.com/schymans/VOM), release v0.5 is used in this study (http://doi.org/10.5281/zenodo.3630081). The full analysis including all scripts and data are available on renku (https://renkulab.io/gitlab/remko.nijzink/vomcases), a static version of this repository can be found on zenodo.org (https://doi.org/10.5281/zenodo.5789101).

*Author contributions.* SJS and RN designed the set-up of the study. Model code was originally developed by SJS, but updated and modified by RN. RN set-up the repositories for the pre-processing, modelling and post-processing on renkulab. LH and JB provided site-specific knowledge and data. The main manuscript and supplementary information were prepared by RN, together with input from SJS. LH and JB provided corrections, suggestions and textual inputs for the main manuscript.

*Competing interests.* The authors declare no conflict of interest

*Acknowledgements.* This study is part of the WAVE-project funded by the Luxembourg National Research Fund (FNR) ATTRACT programme (A16/SR/11254288).

This work used eddy covariance data collected by the TERN-OzFlux facility (http://data.ozflux.org.au/portal/home). OzFlux would like to acknowledge the financial support of the Australian Federal Government via the National Collaborative Research Infrastructure Scheme and the Education Investment Fund.

We acknowledge the SILO Data Drill hosted by the Queensland Department of Environment and Science for providing the meteorological data (https://www.longpaddock.qld.gov.au/silo/).

We acknowledge the Scripps CO2 program (https://scrippsco2.ucsd.edu/data/atmospheric_co2/primary_mlo_co2_record.html) for the Mauna Loa Observatory Records.

We also acknowledge CSIRO for the Soil and Landscape Grid of Australia (https://aclep.csiro.au/aclep/soilandlandscapegrid/index.html) and the Australian monthly fPAR derived from Advanced Very High Resolution Radiometer reflectances - version 5 (https://data.csiro.au/dap/landingpage?pid=csiro:6084).

We acknowledge the Northern Territory Water Data WebPortal for the groundwater data (https://water.nt.gov.au/).

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
