# Peer review of "Influence of modifications (from AoB2015 to v0.5) in the Vegetation Optimality Model"

_Geoscientific Model Development, 2021_

## Referee Comment (RC2)

**General Comments:**

This technical note is a companion paper to a model inter-comparison (https://doi.org/10.5194/hess-2021-265) testing the efficacy of the Vegetation Optimality Model (VOM) to conventional TBMs for five savanna sites on the North Australian Tropical Transect (NATT). In this technical note, the authors perform a sensitivity analysis on the VOM at a single site, Howard Springs, in response to updated model input data, soil hydrology, optimized grass rooting depth, and an imposed free drainage condition required by the HESS companion paper. The authors perform these updates to the Howard Springs VOM originally run in Schymanski et al. (2015) (Sc15 from here on) one-by-one to detail their influence on evapotranspiration (ET), gross primary productivity (GPP), foliage coverage, and soil water dynamics. Overall, the authors found that changes to soil texture (and, thus, water storage) as well as the groundwater levels had the most significant impacts on mean annual GPP and ET.  Overall, ET was substantially less annually (~19%) due to the free drainage conditions, whereas, GPP was essentially the same (~3% change) due to compensating effects from soil texture change and free drainage.

Overall, I commend the authors for taking on the tedious, yet important task of disentangling the effects of model updates on resulting outputs and revealing interesting model sensitivities to soil hydrology that are often taken for granted. I do think this paper can serve as a blueprint for assessing model changes and highlights interesting soil-vegetation dependencies for future study.  Primarily, my comments focus on clarifying the introduction/motivation for this work and bolstering the interpretations of soil hydrology model updates on fluxes with results from the Supplement.

Similar to Reviewer 1, I was confused by the motivation in the introduction.  The authors appear to conflate the motivations for the HESS model inter-comparison and this technical note.  I have provided suggestions for clarifying the relation between the two papers and sharpening the motivation for this technical note. I understand the authors have already addressed some of my comments in their response to Reviewer 1, but I have included them for completeness.

Although this technical note does well to synthesize VOM sensitivity, I thought that the authors' analysis of the effects of soil texture and drainage updates on fluxes ignored significant changes in root water uptake, plant water use strategy, and fractional cover shown in Supplement S1. Furthermore, I feel the authors' under-utilize the large amount of information in Supplement S1 (contains 54 figures but only referenced generically twice). Therefore, I have tried to ask clarifying questions and refer to figures in the supplement that could possibly explain ambiguities in the authors' interpretations.  I hope this helps the authors create a more complete picture of the VOM sensitivity to soil hydrology and highlight interesting results contained in the Supplement.

All other comments relate to clarifying methods, formatting figures, and other technical corrections.  I hope the authors find these comments helpful and I look forward to their responses.

**Specific Comments:**

Lines 6: Instead of saying "… a range of updates to previous applications of the VOM have been made for increased generality and improved comparability with conventional models", you should explicitly reference the HESS companion paper. Then, the following sentence should define the purpose of this technical note in relation to the HESS paper.  I think it is best at the outset to clearly differentiate the

HESS paper and this technical note. As written, it makes me think you are going to perform both the work of the HESS paper and the sensitivity analysis in this technical note.

Line 7: The wording "To assess in how far the updates…." is confusing. I would suggest simplifying it.

Line 30-35: Can you cite either sources for the optimality theory or empirical evidence for some of these assumptions for the interested reader? The assumption that maintenance costs of plant organ functionality are transferrable between species is interesting. Is there any evidence you can cite for this point?

Lines 38-43: I feel these lines could be assimilated into Lines 29-37, where you introduce the optimality theory. I think you can condense this and maybe introduce the optimality theory by introducing the VOM. As Reviewer 1 commented, Lines 40-42 are redundant.

Lines 42-53: This text appears to be the motivation for the HESS companion paper and not this technical note. You introduce the NATT sites and TBM issues raised by the Whitley et al. (2016) inter-comparison study (Wh16 from here on). Next, you state your goal is to determine if the VOM can alleviate the Wh16 issues by running the VOM at the different NATT sites with the same conditions as Wh16. This is very confusing motivation and goal for this technical note given: 1) you only use the Howard Springs site and 2) there are no comparisons with the results from Wh16. Here is what I suggest to clarify the motivation for this technical note:

1) Explicitly reference the HESS companion paper and state its goals, which are to see if VOM can address the Wh16 shortcomings. Also, I would more clearly define the Wh16 shortcomings that the HESS paper attempts to address.
2) State the purpose of this technical note in relation to the HESS companion paper.
3) State the motivation for doing this systematic model sensitivity analysis with Sc15? Why is this important? Does this technical note provide additional insights into the conclusions of the HESS companion paper? What are they?

Lines 52-62: These lines seem to lay out the updates applied to the new VOM from Sc15. I think these can be briefly summarized here as they are laid out in detail in the methods.

Lines 63-66: This seems to be closer to the point of this technical note. You must add clearer context motivating this analysis (see my comment on Lines 42-53). Additionally, I feel this paragraph is missing implications. What is the importance of this work for future modelling applications or use of the VOM?

Sect. 2.1: It would be helpful to justify why you picked Howard Springs out of the five NATT sites used in Wh16 and your HESS companion paper.

Sect 2.2: I had a hard time figuring out why some equations were included and others were not. I can certainly understand not wanting to repeat the longer methodologies in Sc15 and other publications; however, I think a few key equations for root water uptake, photosynthesis and soil water transport can give the reader a better feel for the optimized parameters in Table 3. Alternately, you could write a sentence or two at the beginning of the section explaining which equations you are showing and why. I have noted below where I thought additional equations may help.

Line 88: An issue pointed out by Wh16 was the representation of C4 grasses in TBMs. Was the seasonal vegetation represented as C3 or C4 in VOM? If represented as C3, you should probably justify this simplification.

Sect. 2.2.1: For both photosynthesis and root water uptake, it may be helpful to at least include the main equation(s) for each. In particular, the equations that contain the optimizable variables from Table 3 (electron transport rate, root surface area, etc).

Lines 105-106: It may be helpful to the reader to briefly explain the difference between your method and the traditional Cowan and Farquhar method. In line 111, you mention soil water marginal cost; which is obviously different from the water marginal cost.

Lines 113: What is your modeling time step? Hourly? Daily? Here, you mention diurnal variations in $G_s$, but, in Table 3, the time scale is shown to be daily. I would explicitly state the time step for your model somewhere.

Line 116: Do you mean the root systems are adjusted in terms of finding the optimal parameter values for the 30-year simulation or they are dynamically adjusted during the simulation? I assume the former given the following sentence; however, it may be good to clear this up.

Line 120: Photosynthetic capacities and root surface area distributions are vague since the equations for each are not shown in Sect. 2.2.1. It does seem that you wanted to avoid explaining the whole photosynthetic model and root water uptake equations, but it may help the reader to include at least the main equations in Sect 2.2.1. Then, here you can specifically refer to photosynthetic capacities and root surface area by their parameter names in Table 3.

Line 121: Here you are saying these vegetation properties vary on a daily basis. Does this mean the time step is daily? Earlier, you stated stomatal conductance varies sub-daily (Line 113). Please clear up the time step.

Line 123: Does this mean that you run 27 (3x3x3) separate parameter sets for the day and pick the best one? What are the justifications for performing this type of optimization on these parameters? Maybe briefly discussing the results of previous Schymanski papers that introduced the short-term optimization would be helpful.

Line 136: Is it realistic to use citrus plant parameters to represent evergreens? Is the solution very sensitive to this assumption?

Line 159-160: Wherever you say, "for consistency with other model applications (Whitley et al., 2016)", you should instead explicitly state these updates are required for your HESS companion paper.

Line 165-166: As with other comments, I think more details on subsurface and soil evaporative fluxes would help readers understand how the roles of the soil textural changes, free drainage condition and soil evaporation play on flux changes in the new VOM.

Lines 176-177: Here, the input data all seems daily; however, $G_s$ was previously mentioned as sub-daily (line 113). Can you clarify the time step that you use?

Line 198: Can you explicitly define FPC and how it is derived from $M_{a,s}$ and $M_{a,p}$?

Lines 281-283: Wouldn't the perennial vegetation also suffer from reduced root water uptake due to lower K_sat? According to Table 2, deeper layers have much lower K_sat. I can understand increased water storage in the deeper layers due to soil texture changes, but I would think soil-to-root flow resistance also increases with the finer textures. In fact, it appears Figure 1.41b in Supplement S1 shows reduced perennial ET due to K_sat. Can you explain?

Line 292: I believe you mean the dry season. Otherwise, you contradict your previous statements.

Sect. 3.2 title: The title "Resulting Differences" needs some work. This section does state the overall difference in annual ET and GPP for the "new VOM" in the first sentence. Then, the section goes into the mechanisms for said differences with Figs. 5-8. So maybe state in the title what the differences are between?

Lines 304-305: "… for consistence with free drainage conditions in other models" should be changed to reflect that this update is required by your HESS companion paper.

Sect. 3.2: Is there a reason you only focus on the overall differences between mean annual GPP and ET for Sc15 and the new VOM? I do like how you explain the differences by looking at soil water mechanisms. However, a novel part of the VOM is prediction of vegetation properties, whose changes are not really addressed in this section (besides the optimized root depths in Figure 6). Looking at Sects. 2 and 3 of Supplement S1, it would appear the changes implemented in the new VOM caused major changes to plant water use strategies in perennials (through soil water marginal cost and max electron transport rate Fig. S1.52) as well as fractional cover (Fig. S1.49). These seem like interesting and large changes that should be addressed and used to bolster the claims already made in this section (see my comments below). Furthermore, these large changes in plant water use strategy due to seemingly minor hydrological changes (although clearly not!) could provide interesting points for Sect. 4.

Lines 311-313: Figs. 6c and 7 do indeed show increased water storage under the new soil textural updates; however, they do not necessarily show greater water availability to plants (i.e., transpirable water) resulting in higher perennial ET. In Figure 6c, the new VOM rooting depths are over a meter shallower than Sc15, meaning comparing the water content in the top 5 m exaggerates the difference in transpirable water since the new VOM roots do not access much water below 3 m. Fig. 7 attempts to offer more support to your claim as the new VOM deep retention curve is more gradual over a larger range of water content compared to the homogenous assumptions in Sc15. However, this does not tell us that the additional soil water volume in the root zone is transpirable. The effect of soil textural change on transpirable water in the VOM is determined (to the best of my knowledge) by resistances in soil water uptake, plant water use strategy that determines stomatal sensitivity to soil water potential, and fractional coverage (foliage). From Supplement S1, the reductions in K_sat for the new VOM reduced perennial ET (Fig. S141b), indicating higher resistance in root water uptake. However, changes to both soil texture parameters and K_sat seemingly create a more efficient plant water use strategy (S1.38d and S1.42d) with fractional cover (S1.49e). Therefore, the question is, does the increase in perennial ET from soil texture changes result from greater transpirable water availability due to increased soil water storage? Or altered plant water use strategy? Or both? I think this argument needs to be fleshed out further to truly understand the mechanism for the buffering effect of soil texture on perennial ET.

Lines 313-315: I could also make the argument that the potentials in Fig. 6d and e look dissimilar. I like the figure, but I think it requires a bit more discussion. The new VOM obviously has a shallower rooting depth, which concentrates moisture depletion to 3m compared to the deeper dry-down in Sc15. Also, the onset of dryness appears to occur much sooner in each year for the new VOM compared Sc15. As mentioned in my previous comment, the soil textural differences also affect root water uptake and stomatal sensitivity to these potentials. Therefore, it is not apparent that the differences in timings and water potentials are strongly similar in how they affect ET and GPP. As with my previous comment, I think this argument must be fleshed out further using some of your results from Supplement S1, Sections 1.10-1.11.

Lines 325-327: Update accordingly with your response to my comments on Lines 311-315.

Lines 336: You should state that you made this assumption for the companion paper to compare to Wh16. Also, I do think this finding does coincide, at least partially, with the first of Wh16's deficiencies: Water Access and Tree Rooting depth. It may be good to highlight this point.

**Technical corrections:**

Line 82: Place comma between reference 3 and 4.

Sect 2.2.1: You should reference Fig. 2 somewhere in this section.

Line 92-93: "…maintenance respiration, projected cover, and leaf area turnover and maintenance."

All Equations: It may improve readability to add a centered dot (\cdot in latex) to represent multiplication. The subscripts and letters run together, especially in Eqn. 3-5.

Line 143: Should be "cost factor"

Line 149: "…based on a sensitivity analysis for Howard Springs (see also Supplement S2)."

Line 226: Remove "at each site separately."

Figure 3: I would increase the font size of the axis labels and titles. It is very difficult to read when printed out. Also, the range for the y-axis are inconsistent for panels a) and e) compared to the rest. I would find a way to keep consistent range or zoom in on each figure relative to the max difference. For instance, I feel we lose the message of panel b) because the range is so large yet a key conclusion is GPP can change by about 20%.

Figure 3 caption: I would just state mean annual in the first line and then you can remove "mean annual" from all the following sublabel descriptions. Also, in line 4 of the caption, the word "projected" is misspelled.

Fig. 5: Missing x axis labels. Also, I would make the legend label "VOM" to be "new VOM" for consistency with other figures

Fig. 7: In legend, it is more helpful to put the layer depth and not number, e.g., 20 cm instead of layer 1.

Fig. 7 caption: Line 2, you say "multiple red lines", I believe this is a mistake.

---

## Author Comment (AC3)

**Response to Referee #1**

We'd like to thank Referee #1 for the positive and helpful review. The comments of Referee #1 concern valid points that will help to improve out manuscript.  Below, we will address the comments of Referee #1, with the referee comments written in italics.

*However, I felt that the goals/sequence of the analysis were not explicit enough and the presentation of the methods rather unorganized, that is, it was difficult for me to clearly understand why things were done from the beginning, and identify/distinguish what parameterizations are part of the 2015 model versus the new version. It seems important for the goal of this paper that these aspects be as clear as possible.*

We will revise the Methods section, also based on the specific comments, and will describe the model more generally, and the modifications of the model only in Section 2.2.8, to avoid confusion. More specifically, we believe most confusion came from the descriptions about the water balance and the model input data, where both model versions were described. We will generalize these sections and add the specific details to Section 2.2.8. We will also describe specific goals of each modification in the point-wise description in Section 2.2.8.

*The results are clear but I was left thinking at the end that something was missing in the conclusions to broaden the relevance of this paper to the community and comment on the utility and value of doing such a systematic analysis of individual model changes; discuss the robustness of VOM overall; the implications of the findings; put the information in a broader context versus replicating the Whitney 2016 boundary conditions.*

This is an important point raised by the referee and we will add more context to the conclusions. We will clarify that model development very often suffers from lack of transparency about the effects of various improvements when applied in isolation or in combination. Therefore, our one-step-at-a-time benchmark approach is likely of use to the broader modelling community. In addition, our findings show that the common assumption of freely draining conditions in TBMs can have large effects on the simulated fluxes and therefore the potential for groundwater influence needs to be carefully assessed when interpreting the results.

*I have mostly made comments on the presentation, which I hope improve the structure of the paper for an easier read and increase the relevance of the information.*

***Specific comments***
*The word 'step-wise' in the title is poorly chosen. I expected the modification to be done one at a time in a sequence building on each other – but this was not exactly the case – they were just done one-by-one.*

We agree that the word step-wise leads to confusion and decided to abandon the use of it completely.

*The introduction is not specific enough and difficult to follow to take away the important information. What are the shortcomings that are meant to be addressed? What is proposed to address them, why and how? what is the specific outline of boundary conditions that need to be changed? Referencing the companion and it's goals could be helpful to provide more context and understand the relevance of this paper.*

This is also an important point raised by the reviewer. We elaborated in the introduction on the accompanying paper and its goals, and have re-written the mentioned paragraphs, currently starting from line 50. In addition, we have added a paragraph explaining the importance of tracing back the effects of model modifications to previous applications of the same model, in order to maintain generality of a model.

*In section 2.2, it is helpful to mention in the first paragraph that detailed descriptions are in Schymanski et al. 2009, 2015 (and perhaps even mention the few other earlier ones referenced later about specific processes). But then it doesn't seem necessary to constantly repeat "according to Schymanski XX", "after Schymanski XX"; defined as "Schymanski XX" in the rest of the section. To lighten up the rest of the model description, I suggest clearly stating at the beginning of the section that all parameterizations and processes are the same as in the original references, except those explicitly mentioned. This would better highlight what is different and needs to be remembered and relevant here.*

We agree that for readability, it will be better to mention the references once. We will follow the suggestion of the referee, and add one statement in the first part of section 2.2 and remove the redundant references.

*It may be even worth completely separating the description of the original model structure versus the parameterizations (and their rational ) relevant in this paper, that is, having a completely separate description/list of individual modifications versus having multiple changes mixed into eachother and described together as they seem to be here. A clearer structure may need more thought as these are the key aspect in the paper. Maybe even a table of the information in section 2.2.8 would be an effective summary.*

We believe that most confusion came in the section about the water balance (2.2.5) and the model input data (2.2.7), where both set-ups were described. We will generalize these descriptions and move the specific descriptions to the section about the modifications (2.2.8, now 2.2.9). We will follow a similar approach about the model input data, with a generalized description in section 2.2.7, and a description of the changes in section 2.2.8 (now 2.2.9). Hence, we will separate the model description from the modifications, as suggested by the referee.

*It would be helpful to name or number the multiple different model variants in a more systematic/tractable way versus referring to the 'new' model and "Schymanski 2015" or "previous application" versus "here" or "current". And be able to more easily reference the text while looking at the figures for the step modifications.*

We will refer to the previous model version as VOM-AoB2015, and the new model VOM-v0.5 for clarity, and remove the referring as described by the referee.

*It would be helpful if results in 3.1 follow the same sequence as how the cases are presented in the methods and in the figures. Is there a rational for this sequence (can you explain it?) and can it be consistent throughout?*

Originally, this sequence was chosen based on the importance of the change, with the changes that had strong effects at the end. However, we feel that this is not obvious anymore, and will change the sequence in the section as the referee suggests.

***Line-specific comments***

*L26 -28: These are a very general statement maybe be more specific about what are "novel modelling approaches" ; "fluxes"; "vegetation dynamics". What are specific shortcoming that are relevant to the VOM?*

We will change "vegetation dynamics" to "vegetation dynamics, such as vegetation cover or root surfaces" , "fluxes" to "carbon and water fluxes", and replaced "novel model approaches are needed, especially related to vegetation dynamics" with "models with explicit vegetation dynamics are needed" We will also add a paragraph about the accompanying paper (also based on the specific comments), where we also elaborated on the specific shortcomings in TBMs that are relevant for the VOM.

*L28 "therefore, we use here" seems misplaced relative to the broad explanations in the next 2 paragraphs. Maybe just simply state "Optimality theory predicts …"*

Will be changed accordingly.

*L40-42 this is repeating what was stated in the previous paragraph*

We will remove the sentence.

*L50-63 What shortcomings versus what modifications? In the next few paragraphs there seems to be a mix of information that should be in the model descriptions / site description. I suggest structuring more in parallel and in a more explicit outline of what shortcomings or what boundary conditions are addressed and what modifications were required and tested here to address them.*

We will re-write this part of the introduction, also with respect to the general and specific comments of the referee.

*L65 Could be helpful to mention how many steps were taken.*

We will add the number of changes here, and in section 2.2.8 (now 2.2.9) the number of model cases (9 changes, with a reproduction of the VOM-Aob2015, re-optimization of the VOM,-AoB2015 and the final VOM-v0.5).

*L84 "found in" ?*

Will be changed accordingly.

*L91: Here and elsewhere use Net Carbon Profit or NCP consistently rather than defining it multiple times*

We will remove the multiple definitions of NCP and refer to it only as NCP throughout the manuscript.

*Table 1: maybe write out all words like precipitation. potential evaporation; radiation ; delete '.'After aridity ; correct units of net raditation MJ m-2 year-1*

Will be changed accordingly.

*L103: components – plural?*

Will be changed accordingly.

*L114: "essentially" is informal writing*

We will remove "essentially".

*L142: space between "cost" and "factor"*

Will be changed accordingly.

*L173: Maybe separate model input data from evaluation data in different sections?*

We will separate the paragraph now into "Meteorological data" and "Model evaluation data"

---

## Author Comment (AC4)

**Response to Referee #2**

We would like to thank also Referee #2 for the helpful and insightful review. Referee #2 brings forward several valid points that we will improve on in our revised manuscript. Below, we will address the comments of Referee #2, with the referee comments written in italics.

*Similar to Reviewer 1, I was confused by the motivation in the introduction. The authors appear to conflate the motivations for the HESS model inter-comparison and this technical note. I have provided suggestions for clarifying the relation between the two papers and sharpening the motivation for this technical note. I understand the authors have already addressed some of my comments in their response to Reviewer 1, but I have included them for completeness.*

Thank you for this comment. Even though we indeed had already a critical look at our introduction after the comments of Referee #1, this comment of Referee #2 only emphasizes the importance of it. For that reason, we will work on the introduction in order to clearly state the motivation of this technical note in relation to the accompanying paper in HESS. In addition, we will add references about the importance of using benchmark datasets in modelling. At the same time, we will clarify that we find it important to trace back the origin of differences in model outcomes, in order to fully understand and explain new model results.

*Although this technical note does well to synthesize VOM sensitivity, I thought that the authors' analysis of the effects of soil texture and drainage updates on fluxes ignored significant changes in root water uptake, plant water use strategy, and fractional cover shown in Supplement S1. Furthermore, I feel the authors' under-utilize the large amount of information in Supplement S1 (contains 54 figures but only referenced generically twice). Therefore, I have tried to ask clarifying questions and refer to figures in the supplement that could possibly explain ambiguities in the authors' interpretations. I hope this helps the authors create a more complete picture of the VOM sensitivity to soil hydrology and highlight interesting results contained in the Supplement.*

This is an important point, and we will elaborate more on these changes in the main manuscript, as well as the findings in the Supplement. More specific, we will add a paragraph on the changes in vegetation properties. At the same time, in order to address also some of the specific comments, we will do additional runs with the VOM-v0.5 with the water use parameters of VOM-AoB2015.

*All other comments relate to clarifying methods, formatting figures, and other technical corrections. I hope the authors find these comments helpful and I look forward to their responses.*

***Specific comments***
*Lines 6: Instead of saying "… a range of updates to previous applications of the VOM have been made for increased generality and improved comparability with conventional models", you should explicitly reference the HESS companion paper. Then, the following sentence should define the purpose of this technical note in relation to the HESS paper. I think it is best at the outset to clearly differentiate the HESS paper and this technical note. As written, it makes me think you are going to perform both the work of the HESS paper and the sensitivity analysis in this technical note.*

We will change this entire sentence to: "Several updates to previous applications of the VOM have been made for the study in the accompanying paper of Nijzink et al. (2021), where we assess whether optimality theory can alleviate common shortcomings of conventional models, as identified in a

previous model inter-comparison study along the North Australian Tropical Transect (NATT) (Whitley et al. 2016)"

*Line 7: The wording "To assess in how far the updates...." is confusing. I would suggest simplifying it.*

We will change this to: "Therefore, we assess in this technical paper how..."

*Line 30-35: Can you cite either sources for the optimality theory or empirical evidence for some of these assumptions for the interested reader? The assumption that maintenance costs of plant organ functionality are transferrable between species is interesting. Is there any evidence you can cite for this point?*

This theory was actually formulated mainly by the previous papers of Schymanski et al. (2007), Schymanski et al. (2008) and Schymanski et al. (2009). Hence, the references are limited to these studies, and an important goal of the accompanying HESS-paper is also to test this theory and the accompanying assumptions. Of course, other optimality approaches do exist, and we will add some additional references to give some more background and context.

*Lines 38-43: I feel these lines could be assimilated into Lines 29-37, where you introduce the optimality theory. I think you can condense this and maybe introduce the optimality theory by introducing the VOM. As Reviewer 1 commented, Lines 40-42 are redundant.*

We will remove the redundant lines, but decided to keep the introduction of the VOM at the start of this paragraph.

*Lines 42-53: This text appears to be the motivation for the HESS companion paper and not this technical note. You introduce the NATT sites and TBM issues raised by the Whitley et al. (2016) inter-comparison study (Wh16 from here on). Next, you state your goal is to determine if the VOM can alleviate the Wh16 issues by running the VOM at the different NATT sites with the same conditions as Wh16. This is very confusing motivation and goal for this technical note given: 1) you only use the Howard Springs site and 2) there are no comparisons with the results from Wh16. Here is what I suggest to clarify the motivation for this technical note:*
*1) Explicitly reference the HESS companion paper and state its goals, which are to see if VOM can address the Wh16 shortcomings. Also, I would more clearly define the Wh16 shortcomings that the HESS paper attempts to address.*
*2) State the purpose of this technical note in relation to the HESS companion paper.*
*3) State the motivation for doing this systematic model sensitivity analysis with Sc15? Why is this important? Does this technical note provide additional insights into the conclusions of the HESS companion paper? What are they?*

We agree with these suggestions, and will re-write this part of the introduction.

*Lines 52-62: These lines seem to lay out the updates applied to the new VOM from Sc15. I think these can be briefly summarized here as they are laid out in detail in the methods.*

We will re-write these paragraphs, but will actually add more detail as Referee #1 asked for a clear description in the introduction about the current shortcomings and necessary changes to boundary conditions of the previous version of the VOM. We believe this also addresses point number 3) in the previous comment.

*Lines 63-66: This seems to be closer to the point of this technical note. You must add clearer context motivating this analysis (see my comment on Lines 42-53). Additionally, I feel this paragraph is missing implications. What is the importance of this work for future modelling applications or use of the VOM?*

We will add more context in this part of the introduction and will add more about our motivation to do this analysis.

*Sect. 2.1: It would be helpful to justify why you picked Howard Springs out of the five NATT sites used in Wh16 and your HESS companion paper.*

We used Howard Springs as this was used previously by Schymanski et al. (2009) and Schymanski et al. (2015). Especially Schymanski et al. (2009) contained a detailed comparison with the flux tower observations at this site, and Howard Springs is for that reason a good benchmark case for the VOM. We will clarify why Howard Springs was used.

*Sect 2.2: I had a hard time figuring out why some equations were included and others were not. I can certainly understand not wanting to repeat the longer methodologies in Sc15 and other publications; however, I think a few key equations for root water uptake, photosynthesis and soil water transport can give the reader a better feel for the optimized parameters in Table 3. Alternately, you could write a sentence or two at the beginning of the section explaining which equations you are showing and why. I have noted below where I thought additional equations may help.*

Thank you for pointing this out, the selection was too much focused on the HESS-paper. We will review the presentation of the equations for the technical note as suggested. Hence, we will include equations of the electron transport capacity, $CO_2$-assimilation, leaf respiration and the root water uptake.

*Line 88: An issue pointed out by Wh16 was the representation of C4 grasses in TBMs. Was the seasonal vegetation represented as C3 or C4 in VOM? If represented as C3, you should probably justify this simplification.*

The seasonal and perennial vegetation in the VOM were both modelled as C3-plants. We will clarify this, and add a sentence about the consequence of this simplification.

*Sect. 2.2.1: For both photosynthesis and root water uptake, it may be helpful to at least include the main equation(s) for each. In particular, the equations that contain the optimizable variables from Table 3 (electron transport rate, root surface area, etc).*

We will include the equations for the electron transport capacity, the $CO_2$-assimilation, leaf respiration and root water uptake for completeness.

*Lines 105-106: It may be helpful to the reader to briefly explain the difference between your method and the traditional Cowan and Farquhar method. In line 111, you mention soil water marginal cost; which is obviously different from the water marginal cost.*

In fact, there is no difference, we only formulated an additional relationship between the soil water suction heads and the marginal water costs of assimilation λ. Originally, Cowan and Farquhar stated

that λ should be sensitive to soil water, whereas we use here the water suction head. This was originally done by Schymanski et al. (2009) with the expectation that plants would more likely sense the water suction head, instead of the total available water. We will add some sentences to clarify this.

*Lines 113: What is your modeling time step? Hourly? Daily? Here, you mention diurnal variations in G_s, but, in Table 3, the time scale is shown to be daily. I would explicitly state the time step for your model somewhere.*

The VOM runs with hourly meteorological input data, down-sampled from daily if necessary. Here, we used daily data, which was then converted into hourly data by the VOM. The VOM itself has a variable time stepping approach from hourly to sub-hourly, which depends on the states in the model. The length of the sub-hourly time step is set in a way that the states in the model do not change more than 10%. The vegetation properties in Table 3 are adjusted at a daily timescale, i.e. the vegetation property is changed at the end of a day. We will add some more explanation about this in the section "Model optimization", as well as "Short-term optimization".

*Line 116: Do you mean the root systems are adjusted in terms of finding the optimal parameter values for the 30-year simulation or they are dynamically adjusted during the simulation? I assume the former given the following sentence; however, it may be good to clear this up.*

We will rephrase this part. These parameters here are indeed optimized for the full period and kept constant, but the root systems, in terms of fine root surface area, also adjust on a daily time scale in order to satisfy the canopy water demand. Hence, the root depths are optimized for the full period, but the root surface areas are adjusted on a daily time scale.

*Line 120: Photosynthetic capacities and root surface area distributions are vague since the equations for each are not shown in Sect. 2.2.1. It does seem that you wanted to avoid explaining the whole photosynthetic model and root water uptake equations, but it may help the reader to include at least the main equations in Sect 2.2.1. Then, here you can specifically refer to photosynthetic capacities and root surface area by their parameter names in Table 3.*

We will make this more specific, and use the parameter names in Table 3.

*Line 121: Here you are saying these vegetation properties vary on a daily basis. Does this mean the time step is daily? Earlier, you stated stomatal conductance varies sub-daily (Line 113). Please clear up the time step.*

These variables change at the end of a day, the time step of the VOM is variable from hourly to sub-hourly. Only stomatal conductance varies on a hourly time step. We will also clarify this in the section about the short-term optimization.

*Line 123: Does this mean that you run 27 (3x3x3) separate parameter sets for the day and pick the best one? What are the justifications for performing this type of optimization on these parameters? Maybe briefly discussing the results of previous Schymanski papers that introduced the short-term optimization would be helpful.*

We run 9 separate parameter sets indeed. The root surface area distributions are adjusted based on the canopy water demand, and these are therefore not parameterized with three different values. We will clarify this.

*Line 136: Is it realistic to use citrus plant parameters to represent evergreens? Is the solution very sensitive to this assumption?*

We agree that there is no evidence that citrus plant parameters represent evergreens, as also acknowledged originally by Schymanski et al. (2008), but believe this is currently the best assumption we can make. We will add a short sensitivity analysis in the supplementary material regarding this parameter.

*Line 159-160: Wherever you say, "for consistency with other model applications (Whitley et al., 2016)", you should instead explicitly state these updates are required for your HESS companion paper.*

We will change this accordingly.

*Line 165-166: As with other comments, I think more details on subsurface and soil evaporative fluxes would help readers understand how the roles of the soil textural changes, free drainage condition and soil evaporation play on flux changes in the new VOM.*

We will add more details about these processes as well, with several sentences in the section about the water balance. We will explain that the VOM uses a discretization of the Buckingham-Darcy equation for the fluxes between the layers, as well as that the soil evaporation is a function of the soil saturation in the top layer.

*Lines 176-177: Here, the input data all seems daily; however, $G_s$ was previously mentioned as sub-daily (line 113). Can you clarify the time step that you use?*

We used daily meteorological input data, but the VOM runs with time steps of hourly to sub-hourly. Hence, the daily time series are converted into hourly time series by the VOM. We will clarify this in this section too.

*Line 198: Can you explicitly define FPC and how it is derived from $M_{a,s}$ and $M_{a,p}$?*

Here, we take it as the sum of the perennial and seasonal component: $M_{a,s} + M_{a,p}$. We will add this here explicitly.

*Lines 281-283: Wouldn't the perennial vegetation also suffer from reduced root water uptake due to lower $K_{sat}$? According to Table 2, deeper layers have much lower $K_{sat}$. I can understand increased water storage in the deeper layers due to soil texture changes, but I would think soil-to-root flow resistance also increases with the finer textures. In fact, it appears Figure 1.41b in Supplement S1 shows reduced perennial ET due to $K_{sat}$. Can you explain?*

This is indeed true, lower $K_{sat}$ affects both perennial and seasonal soil-to-root flow. However, the effects of a lower $K_{sat}$ seem to be compensated by the increased soil water storage, as a result of the new soil structures with higher field capacity in the deeper layers. Hence, even though resistivities  may increase, these can still be off-set by higher matrix potentials, that drive the flow. However, we agree

with the other comments of the referee, that we need to closely look at the changes in the water use strategies as well.

*Line 292: I believe you mean the dry season. Otherwise, you contradict your previous statements.*

We will correct this.

*Sect 3.2 title: The title "Resulting Differences" needs some work. This section does state the overall difference in annual ET and GPP for the "new VOM" in the first sentence. Then, the section goes into the mechanisms for said differences with Figs. 5-8. So maybe state in the title what the differences are between?*

We will change this to "Comparing VOM-v0.5 and VOM-AoB2015: resulting differences and underlying mechanisms".

*Lines 304-305: "… for consistence with free drainage conditions in other models" should be changed to reflect that this update is required by your HESS companion paper.*

We will change this accordingly.

*Sect. 3.2: Is there a reason you only focus on the overall differences between mean annual GPP and ET for Sc15 and the new VOM? I do like how you explain the differences by looking at soil water mechanisms. However, a novel part of the VOM is prediction of vegetation properties, whose changes are not really addressed in this section (besides the optimized root depths in Figure 6). Looking at Sects. 2 and 3 of Supplement S1, it would appear the changes implemented in the new VOM caused major changes to plant water use strategies in perennials (through soil water marginal cost and max electron transport rate Fig. S1.52) as well as fractional cover (Fig. S1.49). These seem like interesting and large changes that should be addressed and used to bolster the claims already made in this section (see my comments below). Furthermore, these large changes in plant water use strategy due to seemingly minor hydrological changes (although clearly not!) could provide interesting points for Sect. 4.*

This is indeed a good point. We focused on the differences in GPP and ET initially, as ET and GPP are the most rich and reliable from the observed flux tower data. For that reason, these are good for an assessment of how the final model outputs changed, but we found many more interesting aspects on the way, similar as the referee here. We will add an extra paragraph in our results were we look at the changes in vegetation parameters.

*Lines 311-313: Figs. 6c and 7 do indeed show increased water storage under the new soil textural updates; however, they do not necessarily show greater water availability to plants (i.e., transpirable water) resulting in higher perennial ET. In Figure 6c, the new VOM rooting depths are over a meter shallower than Sc15, meaning comparing the water content in the top 5 m exaggerates the difference in transpirable water since the new VOM roots do not access much water below 3 m. Fig. 7 attempts to offer more support to your claim as the new VOM deep retention curve is more gradual over a larger range of water content compared to the homogenous assumptions in Sc15. However, this does not tell us that the additional soil water volume in the root zone is transpirable. The effect of soil textural change on transpirable water in the VOM is determined (to the best of my knowledge) by resistances in soil water uptake, plant water use strategy that determines stomatal sensitivity to soil water potential, and fractional coverage (foliage). From Supplement S1, the reductions in K_sat for the new VOM*

*reduced perennial ET (Fig. S141b), indicating higher resistance in root water uptake. However, changes to both soil texture parameters and K_sat seemingly create a more efficient plant water use strategy (S1.38d and S1.42d) with fractional cover (S1.49e). Therefore, the question is, does the increase in perennial ET from soil texture changes result from greater transpirable water availability due to increased soil water storage? Or altered plant water use strategy? Or both? I think this argument needs to be fleshed out further to truly understand the mechanism for the buffering effect of soil texture on perennial ET.*

Thank you for this comment, it involves fair questions that we will have to pay attention to. In a first step, we will replace figure 6c with the water stored in the root zone, instead of the upper 5 meters. In addition, we ran the VOM-0.5 with the "old" water use parameters now, as can be seen in the figure below. Here, it can be noted that the differences between the VOM-v0.5 (red) and the VOM-v0.5 with the water use parameters of Schymanski et al. (2015) (black), are relatively small for the perennial ET, indicating that the largest effect comes from the new soil structure. Nevertheless, also the water use parameters contribute, as the perennial ET slightly increased for the new VOM-v0.5 during the wet season, whereas the values with the old water use parameters remain lower.

[Figure]

**Figure 1.** *Comparison for Howard Springs from 2001-2006 (subset from 1980-2017) between the results of Schymanski et al. (2015) (green) and the new results that implemented all changes (VOM-v0.5, red) and the VOM-v0.5 with the water use parameters of Schymanski et al. (2015), for a) ET, b) transpiration perennials (trees), c) transpiration seasonals (grasses), d) soil evaporation, e) GPP, f) GPP perennials (trees), g) GPP seasonals (grasses), all smoothed with a moving average of 7 days, and h) projective cover. The daily average quality flags of the fluxtower observations are shown in dashed lines with a value of 100 when a day is completely gap-filled and 1 when it is observed.*

*Lines 313-315: I could also make the argument that the potentials in Fig. 6d and e look dissimilar. I like the figure, but I think it requires a bit more discussion. The new VOM obviously has a shallower rooting depth, which concentrates moisture depletion to 3m compared to the deeper dry-down in Sc15. Also, the onset of dryness appears to occur much sooner in each year for the new VOM compared Sc15. As mentioned in my previous comment, the soil textural differences also affect root water uptake and stomatal sensitivity to these potentials. Therefore, it is not apparent that the differences in timings and water potentials are strongly similar in how they affect ET and GPP. As with my previous comment, I think this argument must be fleshed out further using some of your results from Supplement S1, Sections 1.10-1.11.*

Thank you for this, we agree that there is more in this figure as we discuss. Hence, we will add a more thorough discussion on this figure, as well as a discussion about the effects of the water use strategies, as mentioned in the previous comment, and the missing groundwater table in the new runs.

*Lines 325-327: Update accordingly with your response to my comments on Lines 311-315.*

Will be changed accordingly.

*Lines 336: You should state that you made this assumption for the companion paper to compare to Wh16. Also, I do think this finding does coincide, at least partially, with the first of Wh16's deficiencies: Water Access and Tree Rooting depth. It may be good to highlight this point.*

Will be changed accordingly.

**Technical corrections:**
*Line 82: Place comma between reference 3 and 4.*

Will be changed accordingly.

*Sect 2.2.1: You should reference Fig. 2 somewhere in this section.*

We will add a reference to the first sentence.

*Line 92-93: "...maintenance respiration, projected cover, and leaf area turnover and maintenance."*

We will rephrase this into: "...maintenance respiration, projected cover is linked to foliage turnover and maintenance costs..."
Note that we discuss here 1) photosynthetic capacity is linked to maintenance respiration and 2) projected cover is linked to leaf are turnover and maintenance.

*All Equations: It may improve readability to add a centered dot (\cdot in latex) to represent multiplication. The subscripts and letters run together, especially in Eqn. 3-5.*

We agree and will add these dots for readability.

*Line 143: Should be "cost factor"*

Will be changed accordingly.

*Line 149: "...based on a sensitivity analysis for Howard Springs (see also Supplement S2)."*

We will remove this sentence based on the comments of Referee #1.

*Line 226: Remove "at each site separately."*

Will be changed accordingly.

*Figure 3: I would increase the font size of the axis labels and titles. It is very difficult to read when printed out. Also, the range for the y-axis are inconsistent for panels a) and e) compared to the rest. I would find a way to keep consistent range or zoom in on each figure relative to the max difference. For instance, I feel we lose the message of panel b) because the range is so large yet a key conclusion is GPP can change by about 20%.*

We will increase the font sizes in this figure, and change the lay-out in a way that each panel contains a similar information content. In addition, we decided to split this figure into a figure with the changes in the fluxes, and a figure with the changes in the vegetation parameters (including the water use strategy parameters, related to other comments of the referee).

*Figure 3 caption: I would just state mean annual in the first line and then you can remove "mean annual" from all the following sublabel descriptions. Also, in line 4 of the caption, the word "projected" is misspelled.*

Will be changed accordingly.

*Fig. 5: Missing x axis labels. Also, I would make the legend label "VOM" to be "new VOM" for consistency with other figures*

We will add the x-axis labels and will rename VOM to VOM-v0.5, following the comments of Referee #1.

*Fig. 7: In legend, it is more helpful to put the layer depth and not number, e.g., 20 cm instead of layer 1.*

We will add the layer depth to this figure.

*Fig. 7 caption: Line 2, you say "multiple red lines", I believe this is a mistake.*

Will be corrected accordingly.

---

## Author Response (AR1)

**Response to Referee #1**

We'd like to thank Referee #1 for the positive and helpful review. The comments of Referee #1 concern valid points that will help to improve out manuscript.  Below, we will address the comments of Referee #1, with the referee comments written in italics.

*However, I felt that the goals/sequence of the analysis were not explicit enough and the presentation of the methods rather unorganized, that is, it was difficult for me to clearly understand why things were done from the beginning, and identify/distinguish what parameterizations are part of the 2015 model versus the new version. It seems important for the goal of this paper that these aspects be as clear as possible.*

We revised the Methods section, also based on the specific comments, and described the model more generally, and the modifications of the model only in Section 2.2.9 (previously 2.2.8), to avoid confusion. More specifically, we believe most confusion came from the descriptions about the water balance and the model input data, where both model versions were described. We generalized these sections and added the specific details to Section 2.2.9. We also described the specific goals of each modification in the point-wise description in Section 2.2.8.

*The results are clear but I was left thinking at the end that something was missing in the conclusions to broaden the relevance of this paper to the community and comment on the utility and value of doing such a systematic analysis of individual model changes; discuss the robustness of VOM overall; the implications of the findings; put the information in a broader context versus replicating the Whitney 2016 boundary conditions.*

This is an important point raised by the referee and we added more context to the conclusions. We clarified that model development very often suffers from lack of transparency about the effects of various improvements when applied in isolation or in combination. Therefore, our one-step-at-a-time benchmark approach is likely of use to the broader modelling community. In addition, our findings show that the common assumption of freely draining conditions in TBMs can have large effects on the simulated fluxes and therefore the potential for groundwater influence needs to be carefully assessed when interpreting the results.

*I have mostly made comments on the presentation, which I hope improve the structure of the paper for an easier read and increase the relevance of the information.*

***Specific comments***
*The word 'step-wise' in the title is poorly chosen. I expected the modification to be done one at a time in a sequence building on each other – but this was not exactly the case – they were just done one-by-one.*

We agree that the word step-wise leads to confusion and decided to abandon the use of it completely.

*The introduction is not specific enough and difficult to follow to take away the important information. What are the shortcomings that are meant to be addressed? What is proposed to address them, why and how? what is the specific outline of boundary conditions that need to be changed? Referencing the companion and it's goals could be helpful to provide more context and understand the relevance of this paper.*

This is also an important point raised by the reviewer. We elaborated in the introduction on the accompanying paper and its goals (lines 50-55), and have re-written the mentioned paragraphs, previously starting from line 50. We described the specific boundary conditions that need to be changed and the reasons for it in lines 61-76.In addition, we have added a paragraph to explain the importance of tracing back the effects of model modifications to previous applications of the same model, in order to maintain generality of a model (lines 78-82).

*In section 2.2, it is helpful to mention in the first paragraph that detailed descriptions are in Schymanski et al. 2009, 2015 (and perhaps even mention the few other earlier ones referenced later about specific processes). But then it doesn't seem necessary to constantly repeat "according to Schymanski XX", "after Schymanski XX"; defined as "Schymanski XX" in the rest of the section. To lighten up the rest of the model description, I suggest clearly stating at the beginning of the section that all parameterizations and processes are the same as in the original references, except those explicitly mentioned. This would better highlight what is different and needs to be remembered and relevant here.*

We agree that for readability, it will be better to mention the references once. We followed the suggestion of the referee, and added one statement in the first part of section 2.2 (lines 107-109) and removed the redundant references.

*It may be even worth completely separating the description of the original model structure versus the parameterizations (and their rational ) relevant in this paper, that is, having a completely separate description/list of individual modifications versus having multiple changes mixed into eachother and described together as they seem to be here. A clearer structure may need more thought as these are the key aspect in the paper. Maybe even a table of the information in section 2.2.8 would be an effective summary.*

We believe that most confusion came in the section about the water balance (2.2.5) and the model input data (2.2.7), where both set-ups were described. We generalized the description of the water balance model (2.2.5) and moved the specific descriptions to the section about the modifications (2.2.9). We followed a similar approach about the model input data, with a generalized description of the meteorological data in section 2.2.7 and model evaluation data in 2.2.8. The description of the changes was again moved to section 2.2.9. Hence, we separated the model description from the modifications, as suggested by the referee.

*It would be helpful to name or number the multiple different model variants in a more systematic/tractable way versus referring to the 'new' model and "Schymanski 2015" or "previous application" versus "here" or "current". And be able to more easily reference the text while looking at the figures for the step modifications.*

We refer now to the previous model version as VOM-AoB2015, and the new model VOM-v0.5 for clarity, and removed the referring as described by the referee.

*It would be helpful if results in 3.1 follow the same sequence as how the cases are presented in the methods and in the figures. Is there a rational for this sequence (can you explain it?) and can it be consistent throughout?*

Originally, this sequence was chosen based on the importance of the change, with the changes that had strong effects at the end. However, we feel that this is not obvious anymore, and changed the sequence in section 3.1 as the referee suggests.

***Line-specific comments***
*L26 -28: These are a very general statement maybe be more specific about what are "novel modelling approaches" ; "fluxes"; "vegetation dynamics". What are specific shortcoming that are relevant to the VOM?*

We changed "vegetation dynamics" to "vegetation dynamics, such as vegetation cover or root surfaces" (line 26) , "fluxes" to "carbon and water flux" (line 27), and replaced "novel model approaches are needed, especially related to vegetation dynamics" with "models with explicit vegetation dynamics are needed" (line 28). We also added a paragraph about the accompanying paper (also based on the specific comments), where we also elaborated on the specific shortcomings in TBMs that are relevant for the VOM (lines 50-82).

*L28 "therefore, we use here" seems misplaced relative to the broad explanations in the next 2 paragraphs. Maybe just simply state "Optimality theory predicts …"*

Changed accordingly.

*L40-42 this is repeating what was stated in the previous paragraph*

We removed the sentence.

*L50-63 What shortcomings versus what modifications? In the next few paragraphs there seems to be a mix of information that should be in the model descriptions / site description. I suggest structuring more in parallel and in a more explicit outline of what shortcomings or what boundary conditions are addressed and what modifications were required and tested here to address them.*

We re-wrote this part of the introduction in lines 61-76, also with respect to the general and specific comments of the referee.

*L65 Could be helpful to mention how many steps were taken.*

We added the number of changes here (lines 85), and in section 2.2.9 (previously 2.2.8) the number of model cases (9 changes, with a reproduction of the VOM-Aob2015, re-optimization of the VOM,-AoB2015 and the final VOM-v0.5, see lines 245-321).

*L84 "found in" ?*

Changed accordingly.

*L91: Here and elsewhere use Net Carbon Profit or NCP consistently rather than defining it multiple times*

We removed the multiple definitions of NCP and refer to it only as NCP throughout the manuscript.

*Table 1: maybe write out all words like precipitation. potential evaporation; radiation ;*

*delete '.' After aridity ; correct units of net raditation MJ m-2 year-1*

Changed accordingly.

*L103: components – plural?*

Changed accordingly.

*L114: "essentially" is informal writing*

We removed "essentially".

*L142: space between "cost" and "factor"*

Changed accordingly.

*L173: Maybe separate model input data from evaluation data in different sections?*

We separated the paragraph now into "Meteorological data" and "Model evaluation data".

**Response to Referee #2**

We would like to thank also Referee #2 for the helpful and insightful review. Referee #2 brings forward several valid points that we will improve on in our revised manuscript. Below, we will address the comments of Referee #2, with the referee comments written in italics.

*Similar to Reviewer 1, I was confused by the motivation in the introduction. The authors appear to conflate the motivations for the HESS model inter-comparison and this technical note. I have provided suggestions for clarifying the relation between the two papers and sharpening the motivation for this technical note. I understand the authors have already addressed some of my comments in their response to Reviewer 1, but I have included them for completeness.*

Thank you for this comment. Even though we indeed had already a critical look at our introduction after the comments of Referee #1, this comment of Referee #2 only emphasizes the importance of it. For that reason, we worked on the introduction in order to clearly state the motivation of this technical note in relation to the accompanying paper in HESS (lines 50-76). In addition, we added references about the importance of using benchmark datasets in modelling  At the same time, we clarified that we find it important to trace back the origin of differences in model outcomes, in order to fully understand and explain new model results (lines 78-82).

*Although this technical note does well to synthesize VOM sensitivity, I thought that the authors' analysis of the effects of soil texture and drainage updates on fluxes ignored significant changes in root water uptake, plant water use strategy, and fractional cover shown in Supplement S1. Furthermore, I feel the authors' under-utilize the large amount of information in Supplement S1 (contains 54 figures but only referenced generically twice). Therefore, I have tried to ask clarifying questions and refer to figures in the supplement that could possibly explain ambiguities in the authors' interpretations. I hope this helps the authors create a more complete picture of the VOM sensitivity to soil hydrology and highlight interesting results contained in the Supplement.*

This is an important point, and we elaborated more on these changes in the main manuscript, as well as the findings in the Supplement. More specific, we added a paragraph on the changes in vegetation properties (lines 375-402) and added a new Figure 4 with the changes in vegetation properties. At the same time, in order to address also some of the specific comments, we did additional runs with the VOM-v0.5 with the water use parameters of VOM-AoB2015 (see Supplement S1, Figures S1.56-S1.57).

*All other comments relate to clarifying methods, formatting figures, and other technical corrections. I hope the authors find these comments helpful and I look forward to their responses.*

***Specific comments***
*Lines 6: Instead of saying "… a range of updates to previous applications of the VOM have been made for increased generality and improved comparability with conventional models", you should explicitly reference the HESS companion paper. Then, the following sentence should define the purpose of this technical note in relation to the HESS paper. I think it is best at the outset to clearly differentiate the HESS paper and this technical note. As written, it makes me think you are going to perform both the work of the HESS paper and the sensitivity analysis in this technical note.*

We changed this entire sentence to: "Several updates to previous applications of the VOM have been made for the study in the accompanying paper of Nijzink et al. (2021), where we assess whether optimality theory can alleviate common shortcomings of conventional models, as identified in a previous model inter-comparison study along the North Australian Tropical Transect (NATT) (Whitley et al. 2016)" (lines 4-7).

*Line 7: The wording "To assess in how far the updates...." is confusing. I would suggest simplifying it.*

We changed this to: "Therefore, we assess in this technical paper how…" (line 7).

*Line 30-35: Can you cite either sources for the optimality theory or empirical evidence for some of these assumptions for the interested reader? The assumption that maintenance costs of plant organ functionality are transferrable between species is interesting. Is there any evidence you can cite for this point?*

This theory was actually formulated mainly by the previous papers of Schymanski et al. (2007), Schymanski et al. (2008) and Schymanski et al. (2009). Hence, the references are limited to these studies, and an important goal of the accompanying HESS-paper is also to test this theory and the accompanying assumptions. Of course, other optimality approaches do exist, and we added some additional references to give some more background and context in lines 28-32.

*Lines 38-43: I feel these lines could be assimilated into Lines 29-37, where you introduce the optimality theory. I think you can condense this and maybe introduce the optimality theory by introducing the VOM. As Reviewer 1 commented, Lines 40-42 are redundant.*

We removed the redundant lines, but decided to keep the introduction of the VOM at the start of this paragraph.

*Lines 42-53: This text appears to be the motivation for the HESS companion paper and not this technical note. You introduce the NATT sites and TBM issues raised by the Whitley et al. (2016) inter-comparison study (Wh16 from here on). Next, you state your goal is to determine if the VOM can alleviate the Wh16 issues by running the VOM at the different NATT sites with the same conditions as Wh16. This is very confusing motivation and goal for this technical note given: 1) you only use the Howard Springs site and 2) there are no comparisons with the results from Wh16. Here is what I suggest to clarify the motivation for this technical note:*
*1) Explicitly reference the HESS companion paper and state its goals, which are to see if VOM can address the Wh16 shortcomings. Also, I would more clearly define the Wh16 shortcomings that the HESS paper attempts to address.*
*2) State the purpose of this technical note in relation to the HESS companion paper.*
*3) State the motivation for doing this systematic model sensitivity analysis with Sc15? Why is this important? Does this technical note provide additional insights into the conclusions of the HESS companion paper? What are they?*

We agree with these suggestions, and re-wrote this part of the introduction, with an explicit reference to the accompanying paper in line 53, a description of the necessary changes in boundary conditions (lines 61-76), and more motivation of this analysis (lines 77-82).

*Lines 52-62: These lines seem to lay out the updates applied to the new VOM from Sc15. I think these can be briefly summarized here as they are laid out in detail in the methods.*

We re-wrote these paragraphs, but actually added more detail as Referee #1 asked for a clear description in the introduction about the current shortcomings and necessary changes to boundary conditions of the previous version of the VOM. We believe this also addresses point number 3) in the previous comment. See lines 61-76 in the revised manuscript.

*Lines 63-66: This seems to be closer to the point of this technical note. You must add clearer context motivating this analysis (see my comment on Lines 42-53). Additionally, I feel this paragraph is missing implications. What is the importance of this work for future modelling applications or use of the VOM?*

We added more context in this part of the introduction and added more about our motivation to do this analysis (lines 77-82).

*Sect. 2.1: It would be helpful to justify why you picked Howard Springs out of the five NATT sites used in Wh16 and your HESS companion paper.*

We used Howard Springs as this was used previously by Schymanski et al. (2009) and Schymanski et al. (2015). Especially Schymanski et al. (2009) contained a detailed comparison with the flux tower observations at this site, and Howard Springs is for that reason a good benchmark case for the VOM. We clarified why Howard Springs was used in lines 94-96.

*Sect 2.2: I had a hard time figuring out why some equations were included and others were not. I can certainly understand not wanting to repeat the longer methodologies in Sc15 and other publications; however, I think a few key equations for root water uptake, photosynthesis and soil water transport can give the reader a better feel for the optimized parameters in Table 3. Alternately, you could write a sentence or two at the beginning of the section explaining which equations you are showing and why. I have noted below where I thought additional equations may help.*

Thank you for pointing this out, the selection was too much focused on the HESS-paper. We will review the presentation of the equations for the technical note as suggested. Hence, we included equations of CO2-assimilation, the electron transport rate and capacity, leaf respiration and the root water uptake (Equations 1, 2, 4, 3 and 5, respectively).

*Line 88: An issue pointed out by Wh16 was the representation of C4 grasses in TBMs. Was the seasonal vegetation represented as C3 or C4 in VOM? If represented as C3, you should probably justify this simplification.*

The seasonal and perennial vegetation in the VOM were both modelled as C3-plants. We clarified this, and added a sentence about the consequence of this simplification (lines 113-115).

*Sect. 2.2.1: For both photosynthesis and root water uptake, it may be helpful to at least include the main equation(s) for each. In particular, the equations that contain the optimizable variables from Table 3 (electron transport rate, root surface area, etc).*

We included the equations for the CO2-assimilation, the electron transport rate and capacity, leaf respiration and root water uptake for completeness (Equations 1, 2, 4, 3 and 5, respectively).

*Lines 105-106: It may be helpful to the reader to briefly explain the difference between your method*

*and the traditional Cowan and Farquhar method. In line 111, you mention soil water marginal cost; which is obviously different from the water marginal cost.*

In fact, there is no difference, we only formulated an additional relationship between the soil water suction heads and the marginal water costs of assimilation λ. Originally, Cowan and Farquhar stated that λ should be sensitive to soil water, whereas we use here the water suction head. This was originally done by Schymanski et al. (2009) with the expectation that plants would more likely sense the water suction head, instead of the total available water. We added some sentences to clarify this (lines 153-157).

*Lines 113: What is your modeling time step? Hourly? Daily? Here, you mention diurnal variations in G_s, but, in Table 3, the time scale is shown to be daily. I would explicitly state the time step for your model somewhere.*

The VOM runs with hourly meteorological input data, down-sampled from daily if necessary. Here, we used daily data, which was then converted into hourly data by the VOM. The VOM itself has a variable time stepping approach from hourly to sub-hourly, which depends on the states in the model. The length of the sub-hourly time step is set in a way that the states in the model do not change more than 10%. The vegetation properties in Table 3 are adjusted at a daily timescale, i.e. the vegetation property is changed at the end of a day. We added more explanation about this in the section "Model optimization", as well as "Short-term optimization".

*Line 116: Do you mean the root systems are adjusted in terms of finding the optimal parameter values for the 30-year simulation or they are dynamically adjusted during the simulation? I assume the former given the following sentence; however, it may be good to clear this up.*

We rephrased this part. These parameters here are indeed optimized for the full period and kept constant, but the root systems, in terms of fine root surface area, also adjust on a daily time scale in order to satisfy the canopy water demand. Hence, the root depths are optimized for the full period, but the root surface areas are adjusted on a daily time scale. We added more explanation about the adjustment of the root surface area in section 2.2.3.

*Line 120: Photosynthetic capacities and root surface area distributions are vague since the equations for each are not shown in Sect. 2.2.1. It does seem that you wanted to avoid explaining the whole photosynthetic model and root water uptake equations, but it may help the reader to include at least the main equations in Sect 2.2.1. Then, here you can specifically refer to photosynthetic capacities and root surface area by their parameter names in Table 3.*

We made this more specific, and used the parameter names in Table 3 (see lines 164-166).

*Line 121: Here you are saying these vegetation properties vary on a daily basis. Does this mean the time step is daily? Earlier, you stated stomatal conductance varies sub-daily (Line 113). Please clear up the time step.*

These variables change at the end of a day, the time step of the VOM is variable from hourly to sub-hourly. Only stomatal conductance varies on a hourly time step. We clarified this in the section about the short-term optimization (2.2.3).

*Line 123: Does this mean that you run 27 (3x3x3) separate parameter sets for the day and pick the best*

*one? What are the justifications for performing this type of optimization on these parameters? Maybe briefly discussing the results of previous Schymanski papers that introduced the short-term optimization would be helpful.*

We run 9 separate parameter sets indeed. The root surface area distributions are adjusted based on the canopy water demand, and these are therefore not parameterized with three different values. We clarified this in this section.

*Line 136: Is it realistic to use citrus plant parameters to represent evergreens? Is the solution very sensitive to this assumption?*

We agree that there is no evidence that citrus plant parameters represent evergreens, as also acknowledged originally by Schymanski et al. (2008), but believe this is currently the best assumption we can make. We added a short sensitivity analysis in the supplementary material regarding this parameter (Supplement S5).

*Line 159-160: Wherever you say, "for consistency with other model applications (Whitley et al., 2016)", you should instead explicitly state these updates are required for your HESS companion paper.*

Changed accordingly.

*Line 165-166: As with other comments, I think more details on subsurface and soil evaporative fluxes would help readers understand how the roles of the soil textural changes, free drainage condition and soil evaporation play on flux changes in the new VOM.*

We added more details about these processes as well, with several sentences in the section about the water balance. We explained here that the VOM uses a discretization of the Buckingham-Darcy equation for the fluxes between the layers (lines 194-197), as well as that the soil evaporation is a function of the soil saturation in the top layer (lines 207-208).

*Lines 176-177: Here, the input data all seems daily; however, G_s was previously mentioned as sub-daily (line 113). Can you clarify the time step that you use?*

We used daily meteorological input data, but the VOM runs with time steps of hourly to sub-hourly. Hence, the daily time series are converted into hourly time series by the VOM. We clarified this in this section too.

*Line 198: Can you explicitly define FPC and how it is derived from M_a,s and M_a,p?*

Here, we take it as the sum of the perennial and seasonal component: M_a,s + M_a,p. We added this here explicitly now (lines 238).

*Lines 281-283: Wouldn't the perennial vegetation also suffer from reduced root water uptake due to lower K_sat? According to Table 2, deeper layers have much lower K_sat. I can understand increased water storage in the deeper layers due to soil texture changes, but I would think soil-to-root flow resistance also increases with the finer textures. In fact, it appears Figure 1.41b in Supplement S1 shows reduced perennial ET due to K_sat. Can you explain?*

This is indeed true, lower K_sat affects both perennial and seasonal soil-to-root flow. However, the effects of a lower K_sat seem to be compensated by the increased soil water storage, as a result of the new soil structures with higher field capacity in the deeper layers. Hence, even though resistivities may increase, these can still be off-set by higher matrix potentials, that drive the flow. However, we agree with the other comments of the referee, that we need to closely look at the changes in the water use strategies as well.

*Line 292: I believe you mean the dry season. Otherwise, you contradict your previous statements.*

We corrected this.

*Sect 3.2 title: The title "Resulting Differences" needs some work. This section does state the overall difference in annual ET and GPP for the "new VOM" in the first sentence. Then, the section goes into the mechanisms for said differences with Figs. 5-8. So maybe state in the title what the differences are between?*

We changed this to "Comparing VOM-v0.5 and VOM-AoB2015: resulting differences and underlying mechanisms".

*Lines 304-305: "... for consistence with free drainage conditions in other models" should be changed to reflect that this update is required by your HESS companion paper.*

We changed this accordingly.

*Sect. 3.2: Is there a reason you only focus on the overall differences between mean annual GPP and ET for Sc15 and the new VOM? I do like how you explain the differences by looking at soil water mechanisms. However, a novel part of the VOM is prediction of vegetation properties, whose changes are not really addressed in this section (besides the optimized root depths in Figure 6). Looking at Sects. 2 and 3 of Supplement S1, it would appear the changes implemented in the new VOM caused major changes to plant water use strategies in perennials (through soil water marginal cost and max electron transport rate Fig. S1.52) as well as fractional cover (Fig. S1.49). These seem like interesting and large changes that should be addressed and used to bolster the claims already made in this section (see my comments below). Furthermore, these large changes in plant water use strategy due to seemingly minor hydrological changes (although clearly not!) could provide interesting points for Sect. 4.*

This is indeed a good point. We focused on the differences in GPP and ET initially, as ET and GPP are the most rich and reliable from the observed flux tower data. For that reason, these are good for an assessment of how the final model outputs changed, but we found many more interesting aspects on the way, similar as the referee here. We added an extra paragraph in our results were we look at the changes in vegetation parameters (lines 375-402) and added a new Figure 4.

*Lines 311-313: Figs. 6c and 7 do indeed show increased water storage under the new soil textural updates; however, they do not necessarily show greater water availability to plants (i.e., transpirable water) resulting in higher perennial ET. In Figure 6c, the new VOM rooting depths are over a meter shallower than Sc15, meaning comparing the water content in the top 5 m exaggerates the difference in transpirable water since the new VOM roots do not access much water below 3 m. Fig. 7 attempts to offer more support to your claim as the new VOM deep retention curve is more gradual over a larger range of water content compared to the homogenous assumptions in Sc15. However, this does not tell*

*us that the additional soil water volume in the root zone is transpirable. The effect of soil textural change on transpirable water in the VOM is determined (to the best of my knowledge) by resistances in soil water uptake, plant water use strategy that determines stomatal sensitivity to soil water potential, and fractional coverage (foliage). From Supplement S1, the reductions in K_sat for the new VOM reduced perennial ET (Fig. S141b), indicating higher resistance in root water uptake. However, changes to both soil texture parameters and K_sat seemingly create a more efficient plant water use strategy (S1.38d and S1.42d) with fractional cover (S1.49e). Therefore, the question is, does the increase in perennial ET from soil texture changes result from greater transpirable water availability due to increased soil water storage? Or altered plant water use strategy? Or both? I think this argument needs to be fleshed out further to truly understand the mechanism for the buffering effect of soil texture on perennial ET.*

Thank you for this comment, it involves fair questions that we will have to pay attention to. In a first step, we replaced figure 6c with the water stored in the root zone, instead of the upper 5 meters. In addition, we ran the VOM-0.5 with the "old" water use parameters now, as can be seen in the figure below. Here, it can be noted that the differences between the VOM-v0.5 (red) and the VOM-v0.5 with the water use parameters of Schymanski et al. (2015) (black), are relatively small for the perennial ET, indicating that the largest effect comes from the new soil structure. Nevertheless, also the water use parameters contribute, as the perennial ET slightly increased for the new VOM-v0.5 during the wet season, whereas the values with the old water use parameters remain lower.

[Figure]

**Figure 1.** *Comparison for Howard Springs from 2001-2006 (subset from 1980-2017) between the results of Schymanski et al. (2015) (green) and the new results that implemented all changes (VOM-v0.5, red) and the VOM-v0.5 with the water use*

*parameters of Schymanski et al. (2015), for a) ET, b) transpiration perennials (trees), c) transpiration seasonals (grasses), d) soil evaporation, e) GPP, f) GPP perennials (trees), g) GPP seasonals (grasses), all smoothed with a moving average of 7 days, and h) projective cover. The daily average quality flags of the fluxtower observations are shown in dashed lines with a value of 100 when a day is completely gap-filled and 1 when it is observed.*

*Lines 313-315: I could also make the argument that the potentials in Fig. 6d and e look dissimilar. I like the figure, but I think it requires a bit more discussion. The new VOM obviously has a shallower rooting depth, which concentrates moisture depletion to 3m compared to the deeper dry-down in Sc15. Also, the onset of dryness appears to occur much sooner in each year for the new VOM compared Sc15. As mentioned in my previous comment, the soil textural differences also affect root water uptake and stomatal sensitivity to these potentials. Therefore, it is not apparent that the differences in timings and water potentials are strongly similar in how they affect ET and GPP. As with my previous comment, I think this argument must be fleshed out further using some of your results from Supplement S1, Sections 1.10-1.11.*

Thank you for this, we agree that there is more in this figure as we discuss. Hence, we added a more thorough discussion about this figure, as well as a discussion about the effects of the water use strategies, as mentioned in the previous comment, and the missing groundwater table in the new runs. See lines 361-364, 418-425.

*Lines 325-327: Update accordingly with your response to my comments on Lines 311-315.*

Changed accordingly.

*Lines 336: You should state that you made this assumption for the companion paper to compare to Wh16. Also, I do think this finding does coincide, at least partially, with the first of Wh16's deficiencies: Water Access and Tree Rooting depth. It may be good to highlight this point.*

Changed accordingly, with an explicit reference to the accompanying paper in line 452, and an additional reference to the findings of Whitley et al. (2016) in lines 449-450.

***Technical corrections:***
*Line 82: Place comma between reference 3 and 4.*

Changed accordingly.

*Sect 2.2.1: You should reference Fig. 2 somewhere in this section.*

We added a reference to the first sentence.

*Line 92-93: "...maintenance respiration, projected cover, and leaf area turnover and maintenance."*

We rephrased this into: "...maintenance respiration, projected cover is linked to foliage turnover and maintenance costs…" (lines 132-135)
Note that we discuss here 1) photosynthetic capacity is linked to maintenance respiration and 2) projected cover is linked to leaf are turnover and maintenance.

*All Equations: It may improve readability to add a centered dot (\cdot in latex) to represent multiplication. The subscripts and letters run together, especially in Eqn. 3-5.*

We agree and added these dots for readability.

*Line 143: Should be "cost factor"*

Changed accordingly.

*Line 149: "...based on a sensitivity analysis for Howard Springs (see also Supplement S2)."*

We removed this sentence based on the comments of Referee #1.

*Line 226: Remove "at each site separately."*

Changed accordingly.

*Figure 3: I would increase the font size of the axis labels and titles. It is very difficult to read when printed out. Also, the range for the y-axis are inconsistent for panels a) and e) compared to the rest. I would find a way to keep consistent range or zoom in on each figure relative to the max difference. For instance, I feel we lose the message of panel b) because the range is so large yet a key conclusion is GPP can change by about 20%.*

We increased the font sizes in this figure, and changed the lay-out in a way that each panel contains a similar information content. In addition, we decided to split this figure into a figure with the changes in the fluxes, and a figure with the changes in the vegetation parameters (including the water use strategy parameters, related to other comments of the referee).

*Figure 3 caption: I would just state mean annual in the first line and then you can remove "mean annual" from all the following sublabel descriptions. Also, in line 4 of the caption, the word "projected" is misspelled.*

Changed accordingly.

*Fig. 5: Missing x axis labels. Also, I would make the legend label "VOM" to be "new VOM" for consistency with other figures*

We added the x-axis labels and will rename VOM to VOM-v0.5, following the comments of Referee #1.

*Fig. 7: In legend, it is more helpful to put the layer depth and not number, e.g., 20 cm instead of layer 1.*

We added the layer depth to this figure.

*Fig. 7 caption: Line 2, you say "multiple red lines", I believe this is a mistake.*

Corrected accordingly.

---

## Referee Report (RR1)

**General comments:**

I compliment the authors on their thorough revisions to the manuscript and thank them for taking the time to run new simulations (see Fig. 4, S1.41, S1.56, S1.57) to answer my previous comments. They have addressed most of my comments thoroughly; however, I do require more clarification on the differing responses of perennial ET and GPP to soil texture (see comments on Lines 357-367 below). In addition, I have included minor grammatical comments and suggestions for making the text easier to read. I recommend this technical note for publication once these minor changes are addressed and look forward to the final product.  Thank you.

**Specific comments:**

Line 8-9: I would replace "original results" with "original analysis of Schymanski et al. (2015)" and adjust the rest of the sentence accordingly.

Line 16: Is this an underestimation of ET and GPP compared to Schymanski et al. (2015) or to the observations? Or both? I think this means compared to Schymanski et al. (2015) as mean annual GPP is overestimated for AoB2015 and v0.5 (Lines 20-21).

Line 20-21: I would specify that GPP is overestimated by 17.8% and 14.7%.

Line 34: "…photosynthesis, water uptake and storage…"

Line 75-76: ", but an evaluation of the effect of this change on the original simulations was not included in Nijzink et al. (2021)." Were any of the effects mentioned in the above paragraphs included in Nijzink et al. (2021)?  If not, I would remove this line.

Line 87: change "works" to "work"

Line 111: Remove "also" from "see also Figure 2"

Line 111-112: Run-on sentence.  Add comma after "(grasses)" or rephrase.

Line 119: Replace "mol fraction" with "mole fraction"

Line 125: I would define what c_Rl is rather than saying it is a constant.  I am guessing it is the cost factor for leaf respiration?

Line 127: I would define h_a and h_d rather than saying they are parameters.

Line 135: I would include the variable name and units after "Root water uptake"

Line 166:  The equations for root surface area distribution are not defined anywhere.  It is easy to imagine how the short term optimization works with M_a,s and J_max,25, but not with S_A,d.  Can the authors either add an equation or discuss how roots are distributed in each layer?

Line 178: Where does the value of 0.22 comes from?

Line 192: Remove "also" from "see also Figure 2"

Line 197: Here and throughout the paper replace "Van Genuchten" with "van Genuchten".

Line 207-208: I am okay with not delving into the full soil evaporation model, but would recommend a parenthetical comment referring to the section of the previous papers that explain the model.

Line 223: I appreciate the authors addressing my previous time-stepping comments. However, this conversion from daily to hourly brought up another question. Is a diurnal variation imposed when converting fluxes like temperature and radiation from daily to hourly? Maybe this is covered in Schymanski et al. (2009), but it may be good to briefly mention here. Otherwise the hourly Cowan and Farquhar calculations would not be very meaningful if the atmospheric conditions were constant over the day.

Line 263: Remove "also" from "see also Supplement S2"

Table 3: Is there a more specific name for the "water use parameters"? The exponent and multiplicative factor have the same description in Table 3.

Sect 3.1: I really like that the authors added further explanation of alterations to vegetation properties under each case as well as Figure 4. I have two (hopefully minor) recommendations for this section to help make it easier to follow:
1) I would indicate the case being discussed in each paragraph to help the reader to attach the text to Table 4. For example, in Line 334 it would be helpful to write "In contrast, changing the fixed atmospheric $CO_2$-levels (350 ppm) in the VOM-AoB2015 to variable atmospheric $CO_2$-levels (Case 2)…"
2) The authors have chosen to discuss the modifications to GPP and ET for each case separately from the modifications to vegetation properties. This can be a little tedious for the reader and may obfuscate the findings. For example, in Lines 335-339, the authors discuss how variable $CO_2$ led to increased GPP, while 10 paragraphs later (Lines 378-382), they discuss how the variable $CO_2$ yields larger perennial vegetation cover. The increase in GPP in the first paragraph is influenced by the vegetation modification in the second paragraph, so it makes sense to combine the two. I would recommend the authors assimilate Lines 375-402 into their respective case paragraphs in Lines 331-374. This will ease the reading by discussing each case completely once as well as make the connection between the modifications of GPP/ET and vegetation properties clearer.

Lines 357-367: I am still struggling with this explanation, which implies that increased vegetation cover and soil storage capacity benefit perennial GPP, while, simultaneously, higher suction head (lower matric potential) and decreased hydraulic conductivity reduce perennial ET. This seems contradictory as stomata are controlling both perennial GPP and ET, which means both fluxes should have a similar response (in sign at least) to changes in cover, soil storage, soil water potential and hydraulic conductivity. The only way different GPP and ET responses make sense, are if water use efficiency and/or photosynthetic capacity per leaf area changes. The authors have illustrated with new analysis (Fig. S1.56-1.57) that the new water use parameters in VOM-v0.5 do not have a large effect on this result. However, I think the answer may lie in the effect of the new soil texture on overall $\lambda_p$ through the higher suction heads. The new soil suctions seem to create more efficient water use in the wet season compared to VOM-AoB2015 (Fig. S1.53d) and could lead to the different sensitivity of GPP and ET to changes in soil texture. To summarize, can the authors explain how perennial GPP and ET can have opposite responses to increased

vegetation cover/soil water capacity and decreased water potential/conductivity when they are both controlled by stomata?

Line 361-363: How does a larger soil moisture capacity and carry-over lead to higher suction heads? Is my understanding that higher suction head means more negative matric potential correct? If so, then wouldn't higher suction heads compound (and not be compensated by) the effects of lower hydraulic conductivity mentioned in Line 364? It appears from Figure S1.40c-e that although there is more water in the soil under the new texture, it will be harder for the roots to extract it due to lower water potentials and less conductivity.  This intuitively explains the reduction in perennial ET even though there is greater vegetative cover, but does not address how there can be greater perennial GPP (see previous comment). Can the author's elaborate?

Lines 372-374: I am not sure the authors meant to make this a new paragraph, but it seems to be part of the previous paragraph.

Sect. 3.2:  I still find the purpose of this section unclear.  Currently, the section compares overall performance of the VOM-v0.5 (Case 12) and VOM-AoB2015 (Case 1) at predicting Howard Springs ET and GPP (Fig. 6). Next, the section explores the mechanisms for the model differences driven primarily by subsurface changes (Fig. 7). This section could be helped by clearly stating that you are comparing VOM-v0.5 (Case 12) used in the companion paper to the VOM-AoB2015 (Case 1) to the Howard Springs data.  Then, let the reader know you are diving further into the model differences by exploring the compensating effects of the two most important factors from Sect. 3.1, soil texture and free drainage. Lastly, the primary focus in this section is on ET, but it would be helpful to say a bit more about GPP effects.

Line 405: I would write "…mean annual GPP changed from 17.8% to 14.7% overestimation." to be consistent with the ET description.

Line 414-415: What is the significance of this difference?  Is it due to the fact you are comparing soil moisture at 5 cm to integrated soil moisture at 20 cm?  I would expect the model to be wetter since deeper soil layers tend to be wetter than near the surface, where transpiration, soil evaporation and loss to deeper layers via gravity and suction occur.

Lines 420-425: This portion will need to be updated based on the response to Lines 361-367.

Line 429: Remove extra parentheses after "(Nijzink et al., 2021)"

Line 437 – 444:  It would be helpful to reference the figures that illustrate these conclusions.

Line 447: Can the authors elaborate on why this effect is more important on highly permeable soils.  I could see tighter soils having a larger capillary fringe and interacting with the root zone significantly.

---

## Author Response (AR2)

**Final response**

We would like to thank the editor and the reviewers for assessing the manuscript. We made the necessary changes following the comments of the reviewers. Below, we first address the comments of Referee #2, followed by our response to Referee #3. In the following, the referee comments are written in italics.

**Response to Referee #2**

*I compliment the authors on their thorough revisions to the manuscript and thank them for taking the time to run new simulations (see Fig. 4, S1.41, S1.56, S1.57) to answer my previous comments. They have addressed most of my comments thoroughly; however, I do require more clarification on the differing responses of perennial ET and GPP to soil texture (see comments on Lines 357-367 below). In addition, I have included minor grammatical comments and suggestions for making the text easier to read. I recommend this technical note for publication once these minor changes are addressed and look forward to the final product. Thank you.*

We are happy the referee valued our efforts to improve the manuscript, and would like to thank the referee also this time for his thorough review.

*Line 8-9: I would replace "original results" with "original analysis of Schymanski et al. (2015)" and adjust the rest of the sentence accordingly.*

We changed this to "original results of Schymanski et al. (2015), and we implemented these changes one at a time".

*Line 16: Is this an underestimation of ET and GPP compared to Schymanski et al. (2015) or to the observations? Or both? I think this means compared to Schymanski et al. (2015) as mean annual GPP is overestimated for AoB2015 and v0.5 (Lines 20-21).*

Indeed, we discuss here in comparison with Schymanski et al. (2015). We changed this accordingly.

*Line 20-21: I would specify that GPP is overestimated by 17.8% and 14.7%.*

We rephrased this to "whereas the relative errors for the mean annual GPP remained similar with an overestimation that changed from 17.8% to 14.7%."

*Line 34: "...photosynthesis, water uptake and storage..."*

Changed accordingly.

*Line 75-76: ", but an evaluation of the effect of this change on the original simulations was not included in Nijzink et al. (2021)." Were any of the effects mentioned in the above paragraphs included in Nijzink et al. (2021)? If not, I would remove this line.*

We removed this line.

*Line 87: change "works" to "work"*

Changed accordingly.

*Line 111: Remove "also" from "see also Figure 2"*

Changed accordingly.

*Line 111-112: Run-on sentence. Add comma after "(grasses)" or rephrase.*

We added a comma.

*Line 119: Replace "mol fraction" with "mole fraction"*

Changed accordingly.

*Line 125: I would define what c_Rl is rather than saying it is a constant. I am guessing it is the cost factor for leaf respiration?*

This factor was defined by Schymanski et al. (2007) and comes from the results of Givnish (1988). This study showed that for a range of species the leaf respiration equals 7% of photosynthetic capacity. We clarified this in the manuscript.

*Line 127: I would define h_a and h_d rather than saying they are parameters.*

We added a description of these parameters, that were originally taken from Medlyn et al. (2002).

*Line 135: I would include the variable name and units after "Root water uptake"*

Changed accordingly.

*Line 166: The equations for root surface area distribution are not defined anywhere. It is easy to imagine how the short term optimization works with M_a,s and J_max,25, but not with S_A,d. Can the authors either add an equation or discuss how roots are distributed in each layer?*

We added the equations and described how the root surface areas are distributed over the layers. Briefly, it is first assessed how far tissue water content gets depleted in the course of the day. This decides whether root surface area will be increased or reduced in the overall profile. In a next step, the effectiveness per layer is determined, i.e. the root water uptake per unit root surface area in each layer. The relative effectiveness of each layer is eventually used to distribute the increases/decreases over the layers, with the biggest increase for the most efficient layer and the largest reduction for the most inefficient layer.

*Line 178: Where does the value of 0.22 comes from?*

This was based on an analysis by Schymanski et al. (2007) of the Glopnet dataset (Wright et al., 2004). We clarified this.

*Line 192: Remove "also" from "see also Figure 2"*

Changed accordingly.

*Line 197: Here and throughout the paper replace "Van Genuchten" with "van Genuchten".*

Changed accordingly.

*Line 207-208: I am okay with not delving into the full soil evaporation model, but would recommend a*

*parenthetical comment referring to the section of the previous papers that explain the model.*

We added a reference to Schymanski et al (2009), including the equation numbers.

*Line 223: I appreciate the authors addressing my previous time-stepping comments. However, this conversion from daily to hourly brought up another question. Is a diurnal variation imposed when converting fluxes like temperature and radiation from daily to hourly? Maybe this is covered in Schymanski et al. (2009), but it may be good to briefly mention here. Otherwise the hourly Cowan and Farquhar calculations would not be very meaningful if the atmospheric conditions were constant over the day.*

We now clarify in Section 2.2.7 that diurnal variation was imposed for global radiation and temperature, and consequently for atmospheric vapour deficit, referring the reader to Appendix A in Schymanski et al. (2009) for details.

*Line 263: Remove "also" from "see also Supplement S2"*

Changed accordingly.

*Table 3: Is there a more specific name for the "water use parameters"? The exponent and multiplicative factor have the same description in Table 3.*

They do not have more specific names, besides defining the non-linearity of the relationship through the exponential factor and the multiplication through the other factor. We changed the names in the table to "exponential water use parameter" and "multiplicative water use parameter" for clarity.

*Sect 3.1: I really like that the authors added further explanation of alterations to vegetation properties under each case as well as Figure 4. I have two (hopefully minor) recommendations for this section to help make it easier to follow:*
*1) I would indicate the case being discussed in each paragraph to help the reader to attach the text to Table 4. For example, in Line 334 it would be helpful to write "In contrast, changing the fixed atmospheric CO2-levels (350 ppm) in the VOM-AoB2015 to variable atmospheric CO2-levels (Case 2)..."*
*2) The authors have chosen to discuss the modifications to GPP and ET for each case separately from the modifications to vegetation properties. This can be a little tedious for the reader and may obfuscate the findings. For example, in Lines 335-339, the authors discuss how variable CO2 led to increased GPP, while 10 paragraphs later (Lines 378-382), they discuss how the variable CO2 yields larger perennial vegetation cover. The increase in GPP in the first paragraph is influenced by the vegetation modification in the second paragraph, so it makes sense to combine the two. I would recommend the authors assimilate Lines 375-402 into their respective case paragraphs in Lines 331-374. This will ease the reading by discussing each case completely once as well as make the connection between the modifications of GPP/ET and vegetation properties clearer.*

Thank you, these are very good suggestions and we made changes accordingly.

*Lines 357-367: I am still struggling with this explanation, which implies that increased vegetation cover and soil storage capacity benefit perennial GPP, while, simultaneously, higher suction head (lower matric potential) and decreased hydraulic conductivity reduce perennial ET. This seems*

*contradictory as stomata are controlling both perennial GPP and ET, which means both fluxes should have a similar response (in sign at least) to changes in cover, soil storage, soil water potential and hydraulic conductivity. The only way different GPP and ET responses make sense, are if water use efficiency and/or photosynthetic capacity per leaf area changes. The authors have illustrated with new analysis (Fig. S1.56-1.57) that the new water use parameters in VOM-v0.5 do not have a large effect on this result. However, I think the answer may lie in the effect of the new soil texture on overall λp through the higher suction heads. The new soil suctions seem to create more efficient water use in the wet season compared to VOM-AoB2015 (Fig. S1.53d) and could lead to the different sensitivity of GPP and ET to changes in soil texture. To summarize, can the authors explain how perennial GPP and ET can have opposite responses to increased vegetation cover/soil water capacity and decreased water potential/conductivity when they are both controlled by stomata?*

Thank you for your constructive comments on this matter, we fully agree that it is counter-intuitive and we want to have a good explanation. Our definition of GPP and ET per unit ground area instead of per leaf area may cause some confusion. On a per leaf area basis, it is correct to assume that changes in ET and GPP should have the same sign. However, if leaf area per ground area increases, the increased light attenuation can result in much greater GPP per ground area for the same or even lower ET, similarly to increasing photosynthetic capacity, as the reviewer noted. In this particular case, the increase in perennial cover is mainly due to improved access to soil moisture during the dry season, but it results in largely increased WUE in the wet season, as light attenuation is increased at the same time as soil moisture access is slightly reduced during the wet season due to higher soil suction and reduced hydraulic conductivity. This results in the different λp-values, as noted by the referee. We clarified this in the revised manuscript.

*Line 361-363: How does a larger soil moisture capacity and carry-over lead to higher suction heads? Is my understanding that higher suction head means more negative matric potential correct? If so, then wouldn't higher suction heads compound (and not be compensated by) the effects of lower hydraulic conductivity mentioned in Line 364? It appears from Figure S1.40c-e that although there is more water in the soil under the new texture, it will be harder for the roots to extract it due to lower water potentials and less conductivity. This intuitively explains the reduction in perennial ET even though there is greater vegetative cover, but does not address how there can be greater perennial GPP (see previous comment). Can the author's elaborate?*

We agree that our formulation was confusing here, and thank the reviewer for pointing it out. Contrary to what our statement implied, it is not that higher storage capacity leads to higher suction heads, but finer texture results in reduced hydraulic conductivity and higher suction heads at relatively high soil water contents, and therefore stronger water holding capacity. Stronger water retention is beneficial for dry season conditions, therefore permitting greater perennial cover, but higher suction heads and reduced conductivity make it harder to suck water out during the wet season, as pointed out by the referee. As explained above, improved dry season conditions increase perennial cover, which increases perennial wet season WUE, resulting in the simulated increase in wet season perennial GPP at slightly reduced ET.

*Lines 372-374: I am not sure the authors meant to make this a new paragraph, but it seems to be part of the previous paragraph.*

This should be part of the previous paragraph indeed, we corrected this.

*Sect. 3.2: I still find the purpose of this section unclear. Currently, the section compares overall performance of the VOM-v0.5 (Case 12) and VOM-AoB2015 (Case 1) at predicting Howard Springs ET and GPP (Fig. 6). Next, the section explores the mechanisms for the model differences driven primarily by subsurface changes (Fig. 7). This section could be helped by clearly stating that you are comparing VOM-v0.5 (Case 12) used in the companion paper to the VOM-AoB2015 (Case 1) to the Howard Springs data. Then, let the reader know you are diving further into the model differences by exploring the compensating effects of the two most important factors from Sect. 3.1, soil texture and free drainage. Lastly, the primary focus in this section is on ET, but it would be helpful to say a bit more about GPP effects.*

We added two introductory sentences here, and elaborated also on GPP effects. However, the strongest influences are by the hydrology and the soil, which is why we focused on these here.

*Line 405: I would write "...mean annual GPP changed from 17.8% to 14.7% overestimation." to be consistent with the ET description.*

Changed accordingly.

*Line 414-415: What is the significance of this difference? Is it due to the fact you are comparing soil moisture at 5 cm to integrated soil moisture at 20 cm? I would expect the model to be wetter since deeper soil layers tend to be wetter than near the surface, where transpiration, soil evaporation and loss to deeper layers via gravity and suction occur.*

Thank you for this comment, this is a good point. We also did not expect that the observed and simulated soil moisture would exactly match, but looked also more at the similarity in dynamics. We rephrased this sentence, and added also the point brought forward by the referee, that the soil moisture is expected to be wetter for the model.

*Lines 420-425: This portion will need to be updated based on the response to Lines 361-367.*

We updated this part.

*Line 429: Remove extra parentheses after "(Nijzink et al., 2021)"*

Changed accordingly.

*Line 437 – 444: It would be helpful to reference the figures that illustrate these conclusions.*

We referenced the figures.

*Line 447: Can the authors elaborate on why this effect is more important on highly permeable soils. I could see tighter soils having a larger capillary fringe and interacting with the root zone significantly.*

We had here the rather permeable soils at Howard Springs in mind, in comparison with the other sites along the North Australian Tropical Transect. However, we believe this statement is not fully supported by what we present in this manuscript, so we decided to remove it.

**Response to Referee #3**

*I am not familiar with the VOM approach. This paper (as well as the accompanying paper in HESS) is not complete enough for understanding this approach. Complete, open-source land surface models able to work at all spatial scales, making use of all available observations (including satellite-derived products) are now available. Why do we need this new approach? The Authors claim that VOM does not need calibration of model parameters but in the end they find that soil water transfer processes are key (not a surprise to me!) and that soil properties need to be described. This means that parameter values have to be prescribed at some stage. Tuning rooting depth is a good example of model parameter tuning. This sounds like a contradiction.*

We are sorry that the referee feels that the manuscript, as well as the accompanying paper in HESS, is not complete enough for understanding the approach. We made changes in the Methods section to improve the clarity, by moving the section about the water balance and the carbon costs before the sections about the optimization. At the same time, we added an extra equation defining the Net Carbon Profit in order to clarify the definition of this objective (Equation 10), which is independent from observations.

The referee argues that existing open-source land surface models work at all spatial scales and make use of all available observations. However, this is exactly the problem that we are pointing out. Using observations of vegetation properties and behaviour as input in these models means that we do not understand how the ecosystem functions, and that making predictions in future scenarios remains highly uncertain. We added some extra sentences in the introduction to underline this.

The VOM reduces the need for observed vegetation properties or behaviour as model input or for model calibration by predicting them based on optimality theory. Consequently, contrary to the reviewer's interpretation, we do not tune rooting depths, but we optimize these for maximizing the Net Carbon Profit, and then compare the simulations with observations and with simulations based on prescribed rooting depths.

*I am extremely concerned by the lack of clarity on how leaf area index (LAI, in m2m-2) is represented. LAI is a key driver of all surface fluxes, including plant transpiration, soil evaporation, rainwater interception. LAI is also related to other quantities like surface albedo. How are LAI and surface albedo represented? How is interception represented? Etc. A Figure showing how these variables compare to observations would be useful. Why not plotting LAI time series in Figures 5 and 6? LAI is mentionned on Line 200 (Eq. 8). But in Eq. 8, a constant "clumped LAI" is used. What is the definition of "clumped LAI"? Why using a constant value of 2.5? Does "clumped LAI" mean "effective LAI"? How do you calculate and validate the clumping index relating true LAI to effective LAI? How and why is Rf from Eq. 8 used in the model?*

The VOM uses a big leaf approach, where LAI is only used to connect the absorbed fraction of PAR with foliage turnover costs. For this purpose, the VOM assumes that a LAI of 2.5 is needed to absorb all the PAR, and that this LAI is reached within the vegetated fraction of a catchment. Since the VOM dynamically predicts the fraction of area covered by vegetation and distinguishes between vegetated and bare soil area fractions, we refer to the LAI within the vegetated area fraction as "clumped LAI", whereas the site-averaged LAI would be that multiplied by the vegetated area fraction. Due to this

simplistic represenation of LAI, we do not present a detailed analysis of the LAI dynamics and per-leaf-area fluxes.  This is also one of our discussion points in the accompanying HESS paper. We clarified in sect 2.2.2 that the LAI is not modelled explicitly and in 2.2.3 we now mention what is meant by clumped LAI and refer to Schymanski et al., 2007 for more details.

Rf is the carbon cost related to  the maintenance of leaf area, we defined this now also in the text. It is used to calculate the NCP, which we defined with the new Equation 10.

Interception was assumed to be negligible at these sites, as there is a strong seasonality with heavy rainfalls. The VOM does not calculate a surface energy balance and hence does not consider the surface albedo.

*Finally, simulations at the site level are presented. Is the VOM able to make 2D simulations? If yes, at which spatial resolution? If not, what would be needed to acquire this capability?*

This is an interesting question. Currently, the VOM only works at the point scale, but a gridded version could be considered in the future. To do so, adjustments to especially the optimization algorithm are necessary, as the vegetation parameters adjust to the local climate and could be different per grid cell. In addition, adjustments to the water balance part may be needed as well, in order to account for the routing of water through the system. However, the focus of this paper is on the processes governing vegetation response to the environment and if/how improvements in process representation propagate into improved model predictions, so we did not expand on the issue of spatially explicit modelling.

---

## Author Response (AR3)

Dear editor,

Thank you for accepting our manuscript. We only added one extra sentence in the acknowledgments, to to thank you and the referees for the feedback.

On behalf of all authors,
Remko Nijzink